# CROSS-DOMAIN REINFORCEMENT LEARNING UNDER DISTINCT STATE-ACTION SPACES VIA HYBRID Q FUNCTIONS

## ABSTRACT

Cross-domain reinforcement learning (CDRL) is meant to improve the data efficiency of RL by leveraging the data samples collected from a source domain to facilitate the learning in a similar target domain. Despite its potential, cross-domain transfer in RL is known to have two fundamental and intertwined challenges: (i) The source and target domains can have distinct state space or action space, and this makes direct transfer infeasible and thereby requires more sophisticated inter-domain mappings; (ii) The domain similarity in RL is not easily identifiable a priori, and hence CDRL can be prone to negative transfer. In this paper, we propose to jointly tackle these two challenges through the lens of hybrid Q functions. Specifically, we propose $Q$Avatar, which combines the Q functions from both the source and target domains with a proper weight decay function. Through this design, we characterize the convergence behavior of $Q$Avatar and thereby show that $Q$Avatar achieves reliable transfer in the sense that it effectively leverages a source-domain Q function for knowledge transfer to the target domain. Through extensive experiments, we demonstrate that $Q$Avatar achieves superior transferability across domains on a variety of RL benchmark tasks, such as locomotion and robot arm manipulation, even in the scenarios of potential negative transfer.

## 1 INTRODUCTION

Reinforcement learning (RL) has witnessed significant progress in various challenging domains, such as game playing (Mnih et al., 2015; Silver et al., 2016), robot control (Gu et al., 2017; Kalashnikov et al., 2018), and language models (Ouyang et al., 2022), mainly due to the integration of general RL techniques with advancements in data collection and computation for large-scale training. However, data inefficiency of RL remains one significant obstacle to its deployment in many real-world applications, where online data collection is either costly (e.g., robotics and autonomous driving) or even hazardous (e.g., medical treatments). As one promising solution, cross-domain RL (CDRL) serves as a practical framework to improve the sample efficiency of RL from the perspective of transfer learning, which leverages the data or the pre-trained models from a source domain to enable knowledge transfer to the target domain, under the presumption that the data collection and model training are much less costly in the source domain (e.g., simulators).

A plethora of the existing CDRL methods focuses on knowledge transfer across environments that share the same state-action spaces but with different transition dynamics. This setting has been extensively studied from a variety of perspectives, such as domain randomization (Peng et al., 2018), learning similarity metrics (Sreenivasan et al., 2023), reward augmentation (Eysenbach et al., 2021; Liu et al., 2022), and data filtering (Xu et al., 2023). Despite the above progress, to fully realize the promise of CDRL, there are two further fundamental challenges to tackle: (i) *Distinct state and/or action spaces between domains*: To support flexible transfer across a wide variety of domains, the generic CDRL algorithms are required to address the discrepancies in the state and action spaces between source and target domains. Take robot control as an example. One common scenario is to apply direct policy transfer across robot agents of different morphologies (Zhang et al., 2021), which naturally leads to discrepancy in representations. This discrepancy significantly complicates the transfer of either data samples or learned source-domain models. (ii) *Unknown domain similarity and negative transfer*: Typical CDRL presumes that the source and target domains are sufficiently

similar such that effective transfer is achievable. However, in practice, given that the data budget of the target domain is limited, it is rather difficult to determine a priori the similarity of a pair of domains, and this becomes even more challenging when the state-action spaces of the two domains are distinct. Moreover, this issue can also be highlighted by the phenomenon of negative transfer (Weiss et al., 2016; Pan & Yang, 2009), where transfer learning from the source domain can have a negative impact on the target domain. As a consequence, despite that CDRL has been shown to succeed in various scenarios, without a proper design, the performance of CDRL could actually be much worse than the vanilla target-domain model learned without using any source knowledge beyond these good-case scenarios. Notably, to tackle (i), several approaches have been proposed to address such representation discrepancy by learning state-action correspondence, either in the typical RL (You et al., 2022) or unsupervised settings (Zhang et al., 2021; Gui et al., 2023). However, these existing solutions are all oblivious to the issues of domain dissimilarity and negative transfer and therefore do not provide any performance guarantees. As a result, one fundamental research question about CDRL remains largely open: *How to achieve efficient and reliable cross-domain transfer in RL across domains of distinct state-action spaces without the knowledge about domain similarity?*

In this paper, we answer the above question in the affirmative. Specifically, we revisit the cross-domain transfer problem in RL from the perspective of *mixing the source-domain and target-domain Q functions* and propose a new CDRL framework termed *QAvatar*, where an "*avatar*", as described in the movie *Avatar*, refers to a genetically engineered body that is created by combining human DNA with the DNA of the native inhabitants of the alien moon. These avatars allow humans on Earth to remotely control these bodies and quickly adapt to the toxic environment of another planet. By drawing an analogy between the cross-planet transfer of humans and the cross-domain transfer of models in RL, we propose to construct a $Q$Avatar, which updates the target-domain policy based on the weighted combination of the learned target-domain Q function and the given source-domain Q function and learn the state-action correspondence by minimizing a cross-domain Bellman loss.

To substantiate this idea, we first present a prototypical algorithm of $Q$Avatar in the tabular setting and establish that $Q$Avatar enjoys a nice upper bound on the sub-optimality under a properly designed weight decay function, regardless of the similarity between the source and target domains. This result also suggests that $Q$Avatar can achieve improved sample efficiency of CDRL while preventing the potential negative transfer. Based on these findings, we further propose a practical implementation by integrating the $Q$Avatar algorithm with a neural mapping function based on a normalizing flow model in learning the state-action correspondence.

The main contributions of this paper can be summarized as follows: 1) We propose the $Q$Avatar framework that achieves knowledge transfer between two domains with distinct state and action spaces for improving sample efficiency. We then present a prototypical $Q$Avatar algorithm and establish its convergence property, showing that $Q$Avatar can improve sample efficiency while avoiding negative transfer. 2) We further substantiate the $Q$Avatar framework by proposing a practical implementation with a normalizing-flow-based state-action mapping. This further demonstrates the compatibility of $Q$Avatar with off-the-shelf methods for learning state-action correspondence. 3) Through extensive experiments and an ablation study, we show that $Q$Avatar significantly outperforms the benchmark CDRL algorithms in various popular RL benchmark tasks, regardless of the quality of source-domain models and domain similarity.

## 2 RELATED WORK

**CDRL across domains with distinct state and action spaces.** The existing approaches can divided into three main categories: (i) *Manually designed latent mapping*: In (Ammar & Taylor, 2012) and (Ammar et al., 2012), the trajectories are mapped manually and by sparse coding from the source domain and the target domain to a common latent space, respectively. The distance between latent states can then be calculated to find the correspondence of the states from the different domains. In Gupta et al. (2017), the correspondence of the states is found by dynamic time warping and the mapping function which can map the states from two domains to the latent space is found by the correspondence. (ii) *Learned inter-domain mapping*: In the literature (Taylor et al., 2008; Zhang et al., 2021; Heng et al., 2022; Gui et al., 2023; Zhu et al., 2024), the inter-domain mapping is mainly learned by enforcing dynamics alignment (or termed dynamics cycle consistency in (Zhang et al., 2021)), i.e., aligning the one-step transitions of the two domains. Additional properties have also

been incorporated as auxiliary loss functions in learning the inter-domain mapping in the prior works, including domain cycle consistency (Zhang et al., 2021; Heng et al., 2022), effect cycle consistency (Zhu et al., 2024), maximizing mutual information between states and embeddings (Heng et al., 2022), and alignment of target-domain rewards with the embeddings (Heng et al., 2022). Moreover, as the state and action spaces are typically bounded sets and these methods directly map the data samples between the two domains, adversarial learning has been used to restrict the output range of the mapping functions (Zhang et al., 2021; Gui et al., 2023). On the other hand, in (Ammar et al., 2015), the state mapping function is found by Unsupervised Manifold Alignment (Wang & Mahadevan, 2009). Despite the above progress, the existing approaches all presume that the domains are sufficiently similar and do not have any performance guarantees (and hence can suffer from negative transfer in bad-case scenarios). By contrast, this paper proposes a robust CDRL method that can achieve transfer regardless of source-domain model quality or domain similarity with guarantees.

**CDRL across domains with identical state and action spaces.** In CDRL, a variety of methods have been proposed for the case where source and target domains share the same state and action spaces but are subject to dynamics mismatch. (i) *Using the data samples from both source and target domains for policy learning*: One popular approach is to use the data from both domains for model updates (Eysenbach et al., 2021; Liu et al., 2022; Xu et al., 2023). For example, for compensating the discrepancy between domains in transition dynamics, (Eysenbach et al., 2021) proposes to modify the reward function, which is learned by an auxiliary domain classifier that distinguishes between the source-domain and target-domain transitions. (Liu et al., 2022) handles the dynamics shift problem in offline RL by augmenting rewards in the source-domain dataset. (Xu et al., 2023) proposes to address dynamics mismatch by a value-guided data filtering scheme, which ensures selective sharing of the source-domain transitions based on the proximity of paired value targets. (ii) *Explicit domain similarity*: (Sreenivasan et al., 2023) proposes to selectively apply direct transfer of the source-domain policy to the target domain based on a learnable similarity metric, which is essentially the TD error of target domain trajectories with source Q function. Moreover, based on the policy invariant explicit shaping (Behboudian et al., 2022), (Sreenivasan et al., 2023) further uses the potential function as a bias term for selecting actions. (iii) *Using both Q-functions for the Q-learning updates*: Target Transfer Q-Learning (Wang et al., 2020) calculates the TD error by the source and target domains Q functions in order to select the TD target from the two Q functions. (iv) *Domain randomization*: To tackle sim-to-real transfer with dynamics mismatch, domain randomization (Rajeswaran et al., 2016; Peng et al., 2018; Chebotar et al., 2019; Du et al., 2021) and Du et al. (2021) collects data from multiple similar source domains with different configurations to learn a high-quality policy that can work robustly in a possibly unseen but similar target domain.

## 3 PRELIMINARIES

In this section, we provide the problem formulation and basic building blocks of CDRL as well as the useful notation needed by subsequent sections. For a set $\mathcal{X}$, we let $\Delta(\mathcal{X})$ denote the set of probability distributions over $\mathcal{X}$. As in typical RL, we model each environment as an infinite-horizon discounted Markov decision process (MDP) denoted by $\mathcal{M} := (\mathcal{S}, \mathcal{A}, P, r, \gamma, \mu)$, where (i) $\mathcal{S}$ and $\mathcal{A}$ represent the state space and action space, (ii) $P : \mathcal{S} \times \mathcal{A} \to \Delta(\mathcal{S})$ denotes the transition function, (iii) $r : \mathcal{S} \times \mathcal{A} \to [-R_{\max}, R_{\max}]$ is the reward function, (iv) $\gamma \in [0, 1)$ is the discounted factor, and (v) $\mu \in \Delta(\mathcal{S} \times \mathcal{A})$ denotes the initial state-action distribution. Notably, the use of an initial distribution over states and actions is a standard setting in the literature of natural policy gradient (NPG) (Agarwal et al., 2021; Ding et al., 2020; Yuan et al., 2022; Agarwal et al., 2020; Zhou et al., 2024). Given any policy $\pi : \mathcal{S} \to \Delta(\mathcal{A})$, we use $\tau = (s_0, a_0, r_1, \cdots)$ to denote a (random) trajectory generated under $\pi$ in $\mathcal{M}$, and the expected total discounted reward under $\pi$ is defined as $V_{\mathcal{M}}^{\pi}(\mu) := \mathbb{E}[\sum_{t=0}^{\infty} \gamma^t r(s_t, a_t)|\pi; s_0, a_0 \sim \mu]$. Moreover, as usual, we use $Q_{\mathcal{M}}^{\pi}(s, a)$ and $V_{\mathcal{M}}^{\pi}(s)$ to denote the Q function and value function of a policy $\pi$. We also define the state-action visitation distribution (also known as the occupancy measure in the MDP literature) of a policy $\pi$ as $d^{\pi}(s, a) := (1 - \gamma)\big(\mu(s, a) + \sum_{t=1}^{\infty} \gamma^t \mathbb{P}(s_t = s, a_t = a; \pi)\big)$, for each $(s, a)$.

**Problem Formulation of Cross-Domain RL.** In typical CDRL, the knowledge transfer involves two MDPs, namely the source-domain MDP $\mathcal{M}_{\text{src}} := (\mathcal{S}_{\text{src}}, \mathcal{A}_{\text{src}}, P_{\text{src}}, r_{\text{src}}, \gamma, \mu_{\text{src}})$ and the target-domain

MDP $\mathcal{M}_{\text{tar}} := (\mathcal{S}_{\text{tar}}, \mathcal{A}_{\text{tar}}, P_{\text{tar}}, r_{\text{tar}}, \gamma, \mu_{\text{tar}})^1$. Notably, in addition to distinct state and action spaces, the two domains can have different reward functions, transition dynamics, and initial distributions. Here we assume that the two MDPs share the same discounted factor $\gamma$, which is rather mild. Moreover, the trajectories of the two domains are completely unpaired. Let $\Pi_{\text{tar}}$ be the set of all stationary Markov policies for $\mathcal{M}_{\text{tar}}$. The goal of the RL agent is to learn a policy $\pi^*$ in the target domain such that the expected total discounted reward is maximized, i.e., $\pi^* := \arg\max_{\pi \in \Pi_{\text{tar}}} V^{\pi}_{\mathcal{M}_{\text{tar}}}(\mu_{\text{tar}})$. To improve sample efficiency via knowledge transfer (compared to learning from scratch), in CDRL, the target-domain agent is granted access to $(\pi_{\text{src}}, Q_{\text{src}}, V_{\text{src}})$, which denotes a policy and the corresponding Q and value functions pre-trained in $\mathcal{M}_{src}$. Notably, we make no assumption on the quality of $\pi_{\text{src}}$ (and hence $\pi_{\text{src}}$ may not be optimal to $\mathcal{M}_{\text{src}}$), despite that $\pi_{\text{src}}$ shall exhibit acceptable performance in practice.

In this paper, we focus on designing a reliable CDRL algorithm in the sense that it effectively leverages a source-domain Q function $Q_{\text{src}}$ for knowledge transfer to the target domain, regardless of the quality of $Q_{\text{src}}$ and domain similarity.

**Inter-Domain Mapping Functions.** To address the discrepancy in state-action spaces in CDRL, learning an inter-domain mapping function is one common building block of many CDRL algorithms. Specifically, there are a variety of ways to construct the mapping functions, such as handcrafted functions (Ammar & Taylor, 2012), encoders and decoders trained by cycle consistency Heng et al. (2022) like cycle-GAN (Zhu et al., 2017), neural networks trained by dynamics alignment of the MDPs (Gui et al., 2023). Moreover, mapping functions have various candidate target spaces, such as a latent space, state or action spaces of the target domain (i.e., from $\mathcal{S}_{\text{src}}, \mathcal{A}_{\text{src}}$ to $\mathcal{S}_{\text{tar}}, \mathcal{A}_{\text{tar}}$), and state or action spaces of the source domain (i.e., from $\mathcal{S}_{\text{tar}}, \mathcal{A}_{\text{tar}}$ to $\mathcal{S}_{\text{src}}, \mathcal{A}_{\text{src}}$). For example, (Gui et al., 2023) proposed to learn two mapping functions $G_1 : \mathcal{S}_{\text{tar}} \to \mathcal{S}_{\text{src}}$ and $G_2 : \mathcal{A}_{\text{src}} \to \mathcal{A}_{\text{tar}}$ through dynamics alignment, which infers the unknown mapping between the unpaired trajectories of $\mathcal{M}_{\text{src}}$ and $\mathcal{M}_{\text{tar}}$ by aligning the one-step state transitions. Specifically, dynamics alignment can be implemented by minimizing the loss function defined as $L(G_1, G_2) = \mathbb{E}_{s_{\text{tar}} \sim \rho, s'_{\text{tar}}, s'_{\text{src}}}\big[\|s'_{\text{src}} - G_1(s'_{\text{tar}})\|_1\big]$, where $s_{\text{tar}}$ is drawn from some target-domain state distribution $\rho$ and $s'_{\text{tar}} \sim P_{\text{tar}}(\cdot | s_{\text{tar}}, G_2(a_{\text{src}}))$ with $a_{\text{src}} \sim \pi_{\text{src}}(\cdot | G_1(s_{\text{tar}}))$. However, this approach provides no performance guarantee as it can suffer from identification issue due to its unsupervised nature. By contrast, in this work, we propose to learn inter-domain state and action mapping functions in the form of $\phi : \mathcal{S}_{\text{tar}} \to \mathcal{S}_{\text{src}}$ and $\psi : \mathcal{A}_{\text{tar}} \to \mathcal{A}_{\text{src}}$ by leveraging a cross-domain Bellman-like loss function with guarantees, as described subsequently in Section 4. Moreover, we construct a toy example to show that dynamics cycle consistency could get stuck at a sub-optimal inter-domain mapping while the proposed cross-domain Bellman-like loss can learn a better mapping by considering the target-domain rewards in Appendix C.1.

**Notation.** Throughout this paper, for any real-valued function $h : \mathcal{S} \times \mathcal{A} \to \mathbb{R}$, for any policy $\pi$, we use $h(s, \pi)$ and $\bar{h}(s, a; \pi)$ as the shorthand for $\mathbb{E}_{a \sim \pi(\cdot|s)}[h(s, a)]$ and $h(s, a) - \mathbb{E}_{a \sim \pi(\cdot|s)}[h(s, a)]$, respectively. For any real vector $z$ and any $p \geq 1$, we use $\|z\|_p$ to denote the $\ell_p$-norm of $z$.

## 4 METHODOLOGY

In this section, we first describe the prototypical framework of $Q$Avatar in the tabular setting (i.e., $\mathcal{S}_{\text{tar}}$ and $\mathcal{A}_{\text{tar}}$ are finite) and establish convergence guarantees. We then extend this framework to a practical deep RL implementation.

### 4.1 THE $Q$AVATAR FRAMEWORK

The main idea of $Q$Avatar is to utilize a weighted combination of a learned target-domain Q function and the given source-domain Q function for robust cross-domain knowledge transfer. In this way, $Q$Avatar can enjoy improved sample efficiency in good-case scenarios (e.g., $\mathcal{M}_{\text{src}}$ and $\mathcal{M}_{\text{tar}}$ are similar) while avoiding potential negative transfer in other scenarios. Specifically, $Q$Avatar consists of the following three major components:

---

[1]Throughout this paper, we use the subscripts "src" and "tar" to represent the objects in the source domain and the target domain, respectively.

---

**Algorithm 1** $Q$Avatar

---

**Require:** Source-domain Q function $Q_{\text{src}}$, weight decay function $\alpha : \mathbb{N} \to [0, 1]$, and $\eta > 0$.
1: Initialize the state mapping function $\phi$, the action mapping function $\psi$, number of on-policy samples per iteration $N_{\text{tar}}$, and the target-domain policy $\pi^{(0)}$
2: **for** iteration $t = 1, \cdots, T$ **do**
3:     Sample $\mathcal{D}_{\text{tar}}^{(t)} = \{(s, a, r, s')\}$ of $N_{\text{tar}}^{(t)}$ on-policy samples using $\pi^{(t)}$ in the target domain
4:     Update $Q_{\text{tar}}$ by minimizing the TD loss in (2), i.e., $Q_{\text{tar}}^{(t)} \leftarrow \arg\min_{Q_{\text{tar}}} \mathcal{L}_{\text{TD}}(Q_{\text{tar}}; \pi^{(t)}, \mathcal{D}_{\text{tar}}^{(t)})$
5:     Update $\phi$ and $\psi$ by minimizing (1), i.e., $\phi^{(t)}, \psi^{(t)} \leftarrow \arg\min_{\phi, \psi} \mathcal{L}_{\text{CD}}(\phi, \psi; Q_{\text{src}}, \pi^{(t)}, \mathcal{D}_{\text{tar}}^{(t)})$.
6:     Update the target-domain policy by adapting NPG to CDRL as in (3).
7: **end for**
8: **Return** Target-domain policy $\pi_{\text{tar}}^{(T)} \sim \text{Uniform}(\{\pi^{(1)}, \cdots, \pi^{(T)}\})$.

---

- **Inter-domain mapping**: Under $Q$Avatar , we propose to learn the inter-domain mappings $\phi : \mathcal{S}_{\text{tar}} \to \mathcal{S}_{\text{src}}$ and $\psi : \mathcal{A}_{\text{tar}} \to \mathcal{A}_{\text{src}}$ by minimizing a cross-domain Bellman-like loss function as

$$\mathcal{L}_{\text{CD}}(\phi, \psi; Q_{\text{src}}, \pi_{\text{tar}}, \mathcal{D}_{\text{tar}}) := \hat{\mathbb{E}}_{(s,a,r_{\text{tar}},s') \in \mathcal{D}_{\text{tar}}} \Big[ \big| r_{\text{tar}} + \gamma \mathbb{E}_{a' \sim \pi_{\text{tar}}}[Q_{\text{src}}(\phi(s'), \psi(a'))] - Q_{\text{src}}(\phi(s), \psi(a)) \big| \Big],$$
(1)

  where $Q_{\text{src}}$ is the pre-trained source-domain Q function and $\mathcal{D}_{\text{tar}} = \{(s, a, r_{\text{tar}}, s')\}$ denotes a set of target-domain samples drawn under $\pi_{\text{tar}}$. Intuitively, the loss in (1) looks for a pair of mapping functions $\phi, \psi$ such that $Q_{\text{src}}$ aligns as much with the target-domain transitions as possible. In the special case of $\mathcal{M}_{\text{src}} = \mathcal{M}_{\text{tar}}$ and $\phi, \psi$ being identity maps, (1) simply reduces to the standard loss function of temporal difference (TD) learning.

- **Target-domain Q function**: To implement the idea of a hybrid Q function, $Q$Avatar maintains a target-domain Q function $Q_{\text{tar}}$, which is essentially a critic of the current target-domain policy. Specifically, in each iteration $t$, $Q_{\text{tar}}$ is obtained by a policy evaluation step via minimizing the standard TD loss for least-squares policy evaluation (LSPE) (Lagoudakis & Parr, 2001; Yu & Bertsekas, 2009; Lazaric et al., 2012)[2], i.e.,

$$\mathcal{L}_{\text{TD}}(Q_{\text{tar}}; \pi_{\text{tar}}, \mathcal{D}_{\text{tar}}) := \hat{\mathbb{E}}_{(s,a,r_{\text{tar}},s') \in \mathcal{D}_{\text{tar}}} \Big[ \big| r_{\text{tar}} + \gamma \mathbb{E}_{a' \sim \pi_{\text{tar}}}[Q_{\text{tar}}(s', a')] - Q_{\text{tar}}(s, a) \big|^2 \Big], \quad (2)$$

  where $\mathcal{D}_{\text{tar}} = \{(s, a, r, s')\}$ denotes target-domain samples.

- **NPG-like policy update with a weighted combination of Q functions**: The core idea of $Q$Avatar is to leverage both $Q_{\text{src}}$ and $Q_{\text{tar}}$ to determine policy updates. In the tabular setting, inspired by (Zhou et al., 2024) in the offline-to-online RL literature, we adapt the classic NPG update (Kakade, 2001), which takes an exponential-weight form on the Q function in the policy space (cf. (Agarwal et al., 2021; Xiao, 2022)), to the CDRL setting. In each iteration $t$,

$$\pi^{(t+1)}(a|s) \propto \pi^{(t)}(a|s) \exp \Big( \eta \cdot \big( (1 - \alpha(t)) Q_{\text{tar}}^{(t)}(s, a) + \alpha(t) Q_{\text{src}}(\phi^{(t)}(s), \psi^{(t)}(a)) \big) \Big), \quad (3)$$

  where $\alpha : \mathbb{N} \to [0, 1]$ is the weight decay function to be configured. Intuitively, $\alpha(t)$ shall be close to one for small $t$ to achieve knowledge transfer from $Q_{\text{src}}$ and gradually diminish to zero to escape from potential negative transfer.

The pseudo code of $Q$Avatar is provided in Algorithm 1.

**Remark 1.** In Line 8 of Algorithm 1, $Q$Avatar outputs the final policy by choosing uniformly at random from the set of all intermediate polices. This is a standard procedure in the optimization literature to connect the average sub-optimality with the performance of output policy. In the experiments, we show that using the last-iterate policy is sufficient and performs well.

## 4.2 PERFORMANCE GUARANTEES OF $Q$AVATAR

In this section, we formally present the theoretical guarantee of $Q$Avatar and thereby describe how to choose the proper decay parameter $\alpha(\cdot)$. Before stating the theorem, we first describe a useful definition on the coverage in terms of state-action distribution (Zhou et al., 2024).

---

[2]These works on LSPE are shown under linear function approximation, which includes the tabular setting as a special case by using one-hot feature vectors.

**Definition 1** (Coverage). *Given a comparator policy $\pi^\dagger$ in $\mathcal{M}_{tar}$, we say that $\pi^\dagger$ has coverage $C_{\pi^\dagger}$ if for any policy $\pi \in \Pi_{tar}$, we have $\|d^{\pi^\dagger}/d^\pi\|_\infty \leq C_{\pi^\dagger}$.*

Notably, one can verify that $C_{\pi^\dagger}$ is finite if $\|d^{\pi^\dagger}/\mu_{\text{tar}}\|_\infty$ is finite (given that $\|\mu_{\text{tar}}/d^\pi\|_\infty \leq 1/(1-\gamma)$ for all $\pi$, by the definition of $d^\pi$), and this can be satisfied under an exploratory initial distribution with $\mu_{\text{tar}}(s,a) > 0$ for all $(s,a)$, which is one standard assumption in the NPG literature (Agarwal et al., 2021; Ding et al., 2020; Yuan et al., 2022; Agarwal et al., 2020; Zhou et al., 2024). Intuitively, the coverage is needed to enable direct comparison of the Bellman error between policies.

**Assumption 1.** *The initial distribution is exploratory, i.e., $\mu_{tar}(s,a) > 0$, for all $s,a$.*

**Definition 2** (TD Error). *For each state-action pair $(s,a)$ and $t \in \mathbb{N}$, the TD error $\epsilon_{td}^{(t)}(s,a)$ is defined as $\epsilon_{td}^{(t)}(s,a) := \left| Q_{tar}^{(t)}(s,a) - r_{tar}(s,a) - \gamma \mathbb{E}_{s' \sim P_{tar}(\cdot|s,a), a' \sim \pi^{(t)}(\cdot|s')}[Q_{tar}^{(t)}(s',a')] \right|$.*

**Definition 3** (Cross-Domain Bellman Error). *Given a source-domain $Q_{src}$, for each state-action pair $(s,a)$ and $t \in \mathbb{N}$, the cross-domain Bellman error $\epsilon_{src,be}^{(t)}(s,a; Q_{src})$ is defined as $\epsilon_{src,be}^{(t)}(s,a; Q_{src}) := \left| Q_{src}(\phi^{(t)}(s), \psi^{(t)}(a)) - r_{tar}(s,a) - \gamma \mathbb{E}_{s' \sim P_{tar}(\cdot|s,a), a' \sim \pi^{(t)}(\cdot|s')}[Q_{src}(\phi^{(t)}(s'), \psi^{(t)}(a'))] \right|$.*

**Definition 4** (Cross-Domain Action Value Function). *For each state-action pair $(s,a)$ and $t \in \mathbb{N}$, the cross-domain action value function $f^{(t)}(s,a)$ is defined as $f^{(t)}(s,a) := (1 - \alpha(t))Q_{tar}^{(t)}(s,a) + \alpha(t)Q_{src}(s,a)$, where $\alpha : \mathbb{N} \to [0,1]$ is the weight decay function.*

Below we use $\|\epsilon_{\text{src, be}}^{(t)}(Q_{\text{src}})\|_\infty$ and $\|\epsilon_{\text{td}}^{(t)}\|_\infty$ as shorthand for $\|\epsilon_{\text{src, be}}^{(t)}(\cdot,\cdot; Q_{\text{src}})\|_\infty$ and $\|\epsilon_{\text{td}}^{(t)}(\cdot,\cdot)\|_\infty$, and we use $\mu_{\text{tar,min}}$ as a shorthand for $\min_{s,a} \mu_{\text{tar}}(s,a)$. We are ready to present the main theoretical result, and the detailed proof is provided in Appendix B.

**Theorem 1.** *(Average Sub-Optimality) Under the QAvatar in Algorithm 1 and Assumption 1, given any fixed learning rate $\eta > 0$, the average sub-optimality over $T$ iterations can be upper bounded as*

$$\frac{1}{T}\sum_{t=1}^{T}\left(V^{\pi^*}(\mu_{tar}) - V^{\pi^{(t)}}(\mu_{tar})\right) \leq \underbrace{\frac{1}{(1-\gamma)T}\sum_{t=1}^{T}\mathbb{E}_{(s,a)\sim d^{\pi^*}}\left[\max_{a'}\bar{f}^{(t)}(s,a')\right] + \frac{\log|\mathcal{A}_{tar}|}{(1-\gamma)T\eta}}_{(a)}$$

$$+ \underbrace{\frac{C_0}{T}\sum_{t=1}^{T}\alpha(t)\|\epsilon_{src,be}^{(t)}(Q_{src})\|_\infty}_{(b)} + \underbrace{\frac{C_0}{T}\sum_{t=1}^{T}(1-\alpha(t))\|\epsilon_{td}^{(t)}\|_\infty}_{(c)},$$

$$\tag{4}$$

*where $C_0 := 2\sqrt{C_{\pi^*}}/((1-\gamma)^2 \mu_{tar,min})$ and $\bar{f}^{(t)}(s,a) := f^{(t)}(s,a) - f^{(t)}(s, \pi^{(t)}(s))$.*

Notably, in (4), the term (a) reflects the learning progress of NPG, the (b) reflects the effect of cross-domain transfer, and (c) indicates the error of policy evaluation for the target-domain policy. The term (c) reflects the sample complexity of the standard least-squares TD-based policy evaluation (Lagoudakis & Parr, 2001; Lagoudakis et al., 2002; Yu & Bertsekas, 2009; Lazaric et al., 2012) and can be made small with sufficient samples (i.e., sufficiently large $N_{\text{tar}}^{(t)}$).

**Key Implications of Theorem 1.**

- **Positive transfer indeed reduces the upper bound of average sub-optimality**: For didactic purposes, consider an ideal case in the sense that $Q_{\text{src}}$ is optimal in the source domain and there always exists a perfect inter-domain mapping $\phi^*$ and $\psi^*$ such that $L_{\text{CD}}(\phi^*, \psi^*; Q_{\text{src}}, \pi_{\text{tar}}, \mathcal{D}_{\text{tar}}) = 0$ under any policy $\pi_{\text{tar}}$. In this case, the positive transfer perfectly happens. Let $\alpha(t)$ be close to one initially in the first $T$ iterations and let $\eta$ be sufficiently large. We can observe that term (b) in (4) is always zero, since $\epsilon_{\text{src, be}}(Q_{\text{src}})$ is always zero. Regarding the term (a) in (4), since $\alpha(t)$ is initially close to one, we have $\bar{f}^{(t)}(s,a) \approx Q_{\text{src}}(\phi^{(t)}(s), \phi^{(t)}(a)) - Q_{\text{src}}(\phi^{(t)}(s), \phi^{(t)}(\pi^{(t)}(s)))$. By the policy update rule in (3), optimality of source Q function $Q_{\text{src}}$, and the fact that perfect inter-domain mapping exists, we know that $\frac{1}{T}\sum_{t=1}^{T}\mathbb{E}_{(s,a)\sim d^{\pi^*}}\left[\max_{a'}\bar{f}^{(t)}(s,a')\right]$ must be small since $\pi(s)$ woul be quickly updated to close to $\pi^*(s)$ for any $s \in \mathcal{S}_{\text{tar}}$. Moreover, the term (c) in (4) shall also be small initially since $\alpha(t)$ is close 1. By combining all the above, we can conclude that the bound on sub-optimality gap is small in the positive transfer regime.

- $Q$**Avatar can avoid getting stuck in the negative transfer regime**: Consider a negative transfer case, where $\mathcal{L}_{\mathrm{CD}}(\phi, \psi; Q_{\mathrm{src}}, \pi_{\mathrm{tar}}, \mathcal{D}_{\mathrm{tar}})$ is always large under any policy $\pi_{\mathrm{tar}}$ and inter-domain mappings $\phi$ and $\psi$. As a result, $\|\epsilon_{\mathrm{src, be}}^{(t)}(Q_{\mathrm{src}})\|_\infty$ is large. In this case, given that $\alpha(t)$ is a decreasing function, $\alpha(t)$ shall be close to 0 under large $t$. We can observe that the term (b) in (4) is small (even if $\|\epsilon_{\mathrm{src, be}}^{(t)}(Q_{\mathrm{src}})\|_\infty$ remains large). Note that the policy update rule in (3) reduces to the original NPG based on the target-domain critic since $\alpha(t)$ is close to 0. For the term (a) in (4), we have $\bar{f}^{(t)}(s, a) \approx Q^{(t)}(s, a) - Q^{(t)}(s, \pi^{(t)}(s))$ and the term (a) in (4) reduces to the standard bound for NPG. Similarly, the term (c) in (4) reduces to the standard TD error since $\alpha(t)$ is close to 0. By combining all the above, we conclude that under $Q$Avatar , the bound on sub-optimality gap would not be continuously dominated by the term (b) in (4) in the negative transfer regime.

**Remark 2.** Note that the proof of Theorem 1 bears some high-level resemblance with (Zhou et al., 2024) as they also use NPG in their hybrid actor-critic (HAC) algorithm. That said, $Q$Avatar is fundamentally different from HAC in two aspects: (i) $Q$Avatar addresses cross-domain transfer while HAC focuses on using offline and online data from the same domain. (ii) $Q$Avatar utilizes the hybrid Q function while HAC applies a hybrid squared error regression loss (i.e., the sum of TD errors calculated from both offline and online data).

### 4.3 PRACTICAL IMPLEMENTATION OF $Q$AVATAR

We extend the $Q$Avatar framework in Algorithm 1 to a practical deep RL implementation for continuous state and action spaces by applying the following design choices. The pseudo code is provided in Algorithm 2 in Appendix.

- **Learning the target-domain policy and the Q function.** To go beyond the tabular setting and handle continuous state and action spaces, we extend $Q$Avatar by first connecting NPG with soft policy iteration (SPI) (Haarnoja et al., 2018). In the entropy-regularized RL setting, SPI has been shown to be a special case of NPG (Cen et al., 2022). Based on this connection, we choose to integrate $Q$Avatar with soft actor-critic (SAC) (Haarnoja et al., 2018), i.e., updating the target-domain critic $Q_{\mathrm{tar}}$ by the critic loss of SAC and updating the target-domain policy $\pi^{(t)}$ by the SAC policy loss function with the weighted combination of $Q_{\mathrm{tar}}$ and $Q_{\mathrm{src}}$ of $Q$Avatar . Regarding the weight decay function $\alpha(t)$, based on the theoretical result, we set $\alpha(t) = t^{-\beta}$ with $\beta > 0$ in the experiments.

- **Learning the inter-domain mapping functions with an augmented flow model.** Similar to the tabular setting, we learn the inter-domain mappings by minimizing the cross-domain Bellman loss. Notably, in practical RL problems, the state and action spaces are mostly bounded sets. As a result, we need to ensure that the outputs of the inter-domain mappings $\phi : \mathcal{S}_{\mathrm{tar}} \to \mathcal{S}_{\mathrm{src}}$ and $\psi : \mathcal{A}_{\mathrm{tar}} \to \mathcal{A}_{\mathrm{src}}$ fall within the feasible regions. As mentioned in Section 2, adversarial learning is widely adopted to solve this practical problem in the existing literature (Taylor et al., 2008; Zhang et al., 2021; Gui et al., 2023; Zhu et al., 2024). However, we observe that adversarial learning could suffer from unstable training process in practice. Therefore, we use the method proposed by (Brahmanage et al., 2024) and train a normalizing flow model that can map the outputs of the mapping functions to the feasible regions.

## 5 EXPERIMENTS

In this section, we show that $Q$Avatar achieves effective cross-domain transfer and improves the sample efficiency on various RL benchmark tasks. Moreover, we demonstrate that $Q$Avatar can still perform well even with the existence of negative transfer between the source and target domains. Unless stated otherwise, all the results reported in this section are averaged over 5 random seeds.

### 5.1 EXPERIMENTAL SETTINGS

**Benchmark CDRL Methods.** We compare the performance of $Q$Avatar with various recent CDRL benchmark algorithms under distinct state-action spaces, including Dynamics Cycle-Consistency (DCC) (Zhang et al., 2021), Cross-Morphology-Domain Policy Adaptation (CMD) (Gui et al., 2023), and Cross-domain Adaptive Transfer (CAT) (Heng et al., 2022). For a fair comparison, all these

benchmark methods and $Q$Avatar use the same set of source-domain models (i.e., the policy and the corresponding Q-networks), which are pre-trained by using SAC in the source domain. However, the original DCC is implemented in a batch setting, i.e., a fixed number of trajectories are collected for learning the inter-domain mappings. For a fair comparison, we adapt the DCC in an online setting, i.e., learning while iteratively collecting new trajectories. Regarding CMD, we observe that the original setting could suffer because the collected trajectories mostly have low returns due to a random behavior policy. Therefore, we consider a stronger version of CMD with target-domain data collected under the target-domain policy, which is induced by the source-domain pre-trained policy and the current inter-domain mappings.

Moreover, to demonstrate the sample efficiency, we also compare $Q$Avatar with the standard SAC (Haarnoja et al., 2018), which learns from scratch in the target domain, as well as with the direct Fine-Tuning (FT) upon the source models (Ha et al., 2024), which can be viewed as using the standard SAC with source feature initialization. Both methods can serve as reasonably competitive baselines. The hyperparameters are provided in Appendix E.

**Evaluation Environments.** We evaluate $Q$Avatar in three types of RL benchmark environments:

- **Locomotion**: We use the standard MuJoCo environments, including Hopper-v3, HalfCheetah-v3 and Ant-v3, as the source domains and follow the same procedure as in (Zhang et al., 2021; Xu et al., 2023) to modify them for the target domains. Moreover, we consider the Centipede environments in CAT (Heng et al., 2022), using CentipedeFour as the source domain and CentipedeSix as the target domain. Additionally, for evaluation in the scenario of negative transfer, we use Inverted Pendulum-v2 and the two-joint robot arm Reacher-v2 as the source domains and employ Inverted Double Pendulum-v2 and the three-joint robot arm Reacher-v2 as the respective target domains. The details about the morphology is in Appendix E.

- **Robot arm manipulation**: We use the environments provided by Robosuite, a popular package for robot learning released by (Zhu et al., 2020). We evaluate our algorithm on three tasks, including block lifting, door opening and table wiping. For each task, we use the Panda robot arm as the source domain and set the UR5e robot arm as the target domain.

- **Autonomous driving**: We use the environments provided by BARK-ML (Bernhard et al., 2020). Notably, these environments are highly unpredictable due to the complex traffic situations and driver behaviors encoded in the BARK-ML behavior model. For cross-domain transfer, we use merging-v0 as the source domain and highway-v0 as the target domain.

Table 1: Dimensionalities of the source and target domains ("Src" and "Tar" represent the source domain and the target domain.

| Environment | State | | Action | |
|---|---|---|---|---|
| | Src | Tar | Src | Tar |
| Hopper | 11 | 13 | 3 | 4 |
| HalfCheetah | 17 | 23 | 6 | 9 |
| Ant | 111 | 133 | 8 | 10 |
| Centipede | 97 | 139 | 10 | 16 |
| IP / Modified IDP | 4 | 11 | 1 | 1 |
| Reacher | 11 | 14 | 2 | 2 |
| Block Lifting | 42 | 47 | 8 | 7 |
| Door Opening | 46 | 51 | 8 | 7 |
| Table Wiping | 37 | 34 | 7 | 6 |
| Merging / Highway | 12 | 16 | 2 | 2 |

Table 1 provides the dimensionalities of the state and action spaces in all the tasks.

**Reproduction and Sanity Checks for DCC, CMD, and CAT.** Regarding CAT and DCC, we directly use the official implementation provided by the original papers. Moreover, as there is no CMD implementation available, we reproduce CMD by referring to the source code of DCC as these two algorithms are similar. As a sanity check, we evaluate DCC and CMD in multiple MuJoCo tasks and confirm that our reproduced scores are indeed close to those reported in the original papers (despite that DCC and CMD do not perform well due to their unsupervised nature). Similarly, a sanity check for CAT on the Centipede task confirms the reproduced score. The details are in Appendix C.2.

### 5.2 Experimental Results

**Does $Q$Avatar improve data efficiency?** As shown by the training curves in Figure 1, we observe that $Q$Avatar achieves improved data efficiency via cross-domain transfer than SAC throughout the training process in all the MuJoCo, Robosuite, and BARK-ML tasks, despite that these tasks have rather different dimensionalities as shown in Table 1. CAT achieves moderate performance

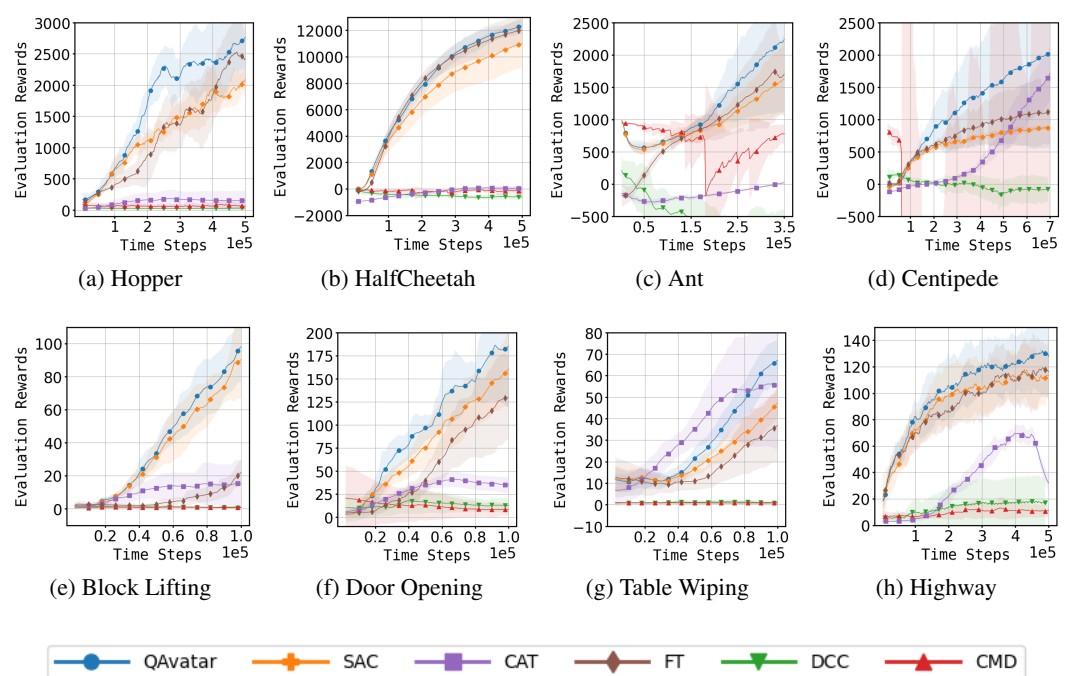

Figure 1: The training curves of $Q$Avatar and the benchmark methods: (a)-(d): Locomotion tasks in MuJoCo; (e)-(g): Robot arm manipulation tasks in Robosuite; (h) Autonomous driving tasks in BARK-ML.

in Table Wiping, Centipede, and Highway but does not learn effectively in the other tasks. These appear reasonable as CAT has no performance guarantees and can suffer if the source and target are rather dissimilar, despite that CAT applies policy gradient with target-domain rewards to align the inter-domain mapping with the target domain. On the other hand, FT typically achieves slight improvement in data efficiency than SAC in MuJoCo but slower learning progress in Robosuite. We conjecture that this is because distinct robot arms in Robosuite lead to more dissimilar state-action representations and hence require more fine-tuning steps.

Regarding CMD, it cannot obtain good returns in most of the tasks. Notably, in some environments like Ant, CMD appears very unstable due to its adversarial learning module for restricting the output of their mapping functions. DCC does not perform well in most tasks, including HalfCheetah. This trend is similar to that in the original paper (Zhang et al., 2021). The rewards obtained by DCC in our experiments are slightly lower than those shown in (Zhang et al., 2021) despite that we try our best to reproduce their results. To strengthen our argument, we offer a comparison of the original and reproduced results in Appendix C.2. We conjecture that the undesired performance of CMD and DCC results from that they learn in an *unsupervised* manner and hence does not take target-domain rewards into account.

Additionally, when we consider the time to threshold metric, our algorithm requires 298k fewer steps to achieve the threshold than SAC does in the best case. When we consider the asymptotic performance metric, our algorithm can obtain higher final rewards than SAC. The results of these two metrics are shown in the Appendix C.3 and C.4.

**Does $Q$Avatar still perform reliably well when negative transfer is likely to happen?** We construct negative transfer scenarios by modifying the environment configurations via swapping action encodings, as described below: (i) In the Reacher environment, we use a two-joint robot arm as the source domain and a three-joint arm as the target domain. To match the action dimensions, the middle joint of the three-joint arm is disabled (i.e., its action is always set to 0). We then alter the three-joint arm's configuration by swapping the encoding of "clockwise" and "counterclockwise" actions (termed "Modified Reacher"). (ii) Similarly, we use Inverted Pendulum (IP) as the source

domain and Inverted Double Pendulum (IDP) as the target domain. Then, we modify the configuration of IDP by swapping the encodings of the actions "left" and "right" (termed "Modified IDP").

As a result, negative transfer shall easily occur if we deactivate the inter-domain action mapping $\psi$ of $Q$Avatar such that $Q$Avatar cannot learn by simply mapping the "clockwise" in three-joint Reacher to "counterclockwise" in two-jointed Reacher and "left" in IDP to "right" in IP. As shown in Figure 2, we observe that $Q$Avatar outperforms all CDRL benchmarks algorithms and exhibits a learning curve similar to SAC and FT, despite the negative transfer scenarios. This confirms that $Q$Avatar can indeed perform reliably due to the use of hybrid Q function.

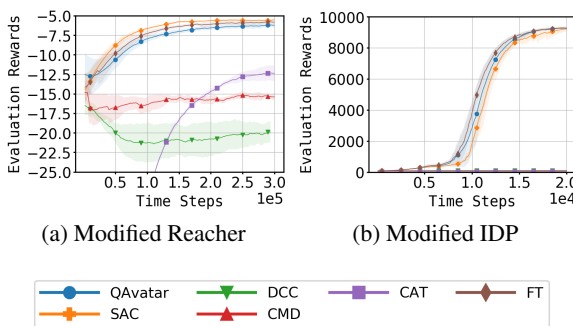

(a) Modified Reacher  (b) Modified IDP

Figure 2: The training curves of $Q$Avatar and the benchmark methods in the negative transfer cases.

**Is $Q$Avatar sensitive to the decay function?** We evaluate $Q$Avatar with $\alpha(t)$ as $1/\sqrt{t}$, $1/t$, and $1/t^{1.5}$. As shown in Figure 3, $Q$Avatar can learn successfully regardless of the choice of $\alpha(t)$ and appear consistently favorable under all these choices of $\alpha(t)$.

**Does $Q$Avatar still perform reliably with a source-domain model of lower quality?** We further run $Q$Avatar with low-quality source-domain Q networks, which are pre-trained only for five thousand steps in both Hopper and Door Opening. As shown by Figure 4, we find that despite $Q$Avatar is affected by the low-quality source model initially, it can quickly catch up and achieve total reward comparable to SAC. This appears consistent with the theoretical result in Theorem 1.

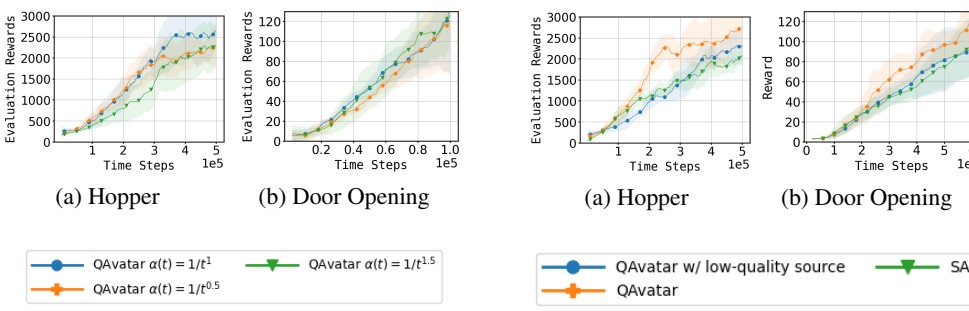

(a) Hopper  (b) Door Opening  (a) Hopper  (b) Door Opening

Figure 3: The training curves of $Q$Avatar under different decay functions $\alpha$.

Figure 4: The training curves of $Q$Avatar with a high-quality and a low-quality source model.

## 6    CONCLUDING REMARKS AND LIMITATIONS

In this paper, we present $Q$Avatar, the first CDRL method that can handle distinct state-action representations between domains with performance guarantees. Based on the idea of combining the source-domain and target-domain Q functions, $Q$Avatar achieves robust knowledge transfer and tackles the negative transfer issue. Through extensive experiments, we show that $Q$Avatar indeed serves as a promising and generic solution to cross-domain transfer in RL. One limitation of this work is that we follow the standard CDRL formulation and consider only one source domain and one target domain. Extending the idea of $Q$Avatar to achieve knowledge transfer from multiple source and target domains is a promising future research direction.

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

## A   SUPPORTING LEMMAS

**Lemma 1** (Performance difference lemma). *For any two policies $\pi$ and $\pi'$, for any state $s$, we have*

$$V^{\pi'}(\mu) - V^{\pi}(\mu) = \frac{1}{1-\gamma} \mathbb{E}_{s,a \sim d^{\pi'}}[A^{\pi}(s,a)],$$

*where $A^{\pi}(s,a) := Q^{\pi}(s,a) - V^{\pi}(s)$ is the advantage function.*

*Proof.* This can be directly obtained from Lemma 6.1 in (Kakade & Langford, 2002). $\qquad \square$

**Lemma 2.** *Suppose $f^{(t)}$ and $\pi^{(t)}$ denote the cross-domain value functions and the policies at iteration $t$. Then, for any learning rate $\eta$ and policy $\pi^*$, we have*

$$\sum_{t=1}^{T} \mathbb{E}_{(s,a) \sim d^{\pi^*}} \left[ f^{(t)}(s,a) - f^{(t)}(s, \pi^{(t)}(s)) \right]$$

$$\leq \sum_{t=1}^{T} \mathbb{E}_{(s,a) \sim d^{\pi^*}} \left[ \max_{a'} f^{(t)}(s,a') - f^{(t)}(s, \pi^{(t)}(s)) \right] + \frac{\log |\mathcal{A}_{tar}|}{\eta}.$$

*Proof.* Let $\bar{f}^{(t)}(s,a) = f^{(t)}(s,a) - f^{(t)}(s, \pi^{(t)}(s))$. According to the policy update rule, at iteration $t$, the policy $\pi^{(t+1)}$ for the next iteration is updated by the formula:

$$\pi^{(t+1)}(a \mid s) = \frac{\pi^{(t)}(a \mid s) \exp\left(\eta \bar{f}^{(t)}(s,a)\right)}{\sum_{a'} \pi^{(t)}(a' \mid s) \exp\left(\eta \bar{f}^{(t)}(s,a')\right)}. \tag{5}$$

Let $Z_t = \sum_{a'} \pi^{(t)}(a' \mid s) \exp\left(\eta \bar{f}^{(t)}(s,a')\right)$. By multiplying both sides of (5) by $Z_t$ taking the logarithm, and then taking the expectation on both sides w.r.t $(s,a) \sim d^{\pi^*}$, we obtain

$$\mathbb{E}_{(s,a) \sim d^{\pi^*}} \left[ \eta \bar{f}^{(t)}(s,a) \right] = \mathbb{E}_{(s,a) \sim d^{\pi^*}} \left[ \log Z_t + \log \pi^{(t+1)}(a \mid s) - \log \pi^{(t)}(a \mid s) \right]. \tag{6}$$

Next, we bound the term $\log Z_t$. Since the $\log(\cdot)$ is an increasing function, we have

$$\log Z_t = \log \left( \sum_{a' \in \mathcal{A}} \pi(a' \mid s) \exp\left(\eta^{(t)} \bar{f}^{(t)}(s,a')\right) \right)$$

$$\leq \log \left( \max_{a' \in \mathcal{A}} \exp\left(\eta \bar{f}^{(t)}(s,a')\right) \right)$$

$$\leq \max_{a' \in \mathcal{A}} \left( \eta \bar{f}^{(t)}(s,a') \right) = \eta \max_{a' \in \mathcal{A}} \bar{f}^{(t)}(s,a').$$

Then, we have

$$\mathbb{E}_{(s,a) \sim d^{\pi^*}} \left[ \eta \bar{f}^{(t)}(s,a) \right] \leq \mathbb{E}_{(s,a) \sim d^{\pi^*}} \left[ \log \pi^{(t+1)}(a \mid s) - \log \pi^{(t)}(a \mid s) + \eta \max_{a'} \bar{f}^{(t)}(s,a') \right]. \tag{7}$$

By taking the summation over iterations on both side of (7),

$$\sum_{t=1}^{T} \mathbb{E}_{(s,a) \sim d^*} \left[ \eta \bar{f}^{(t)}(s,a) \right]$$

$$\leq \sum_{t=1}^{T} \mathbb{E}_{(s,a) \sim d^{\pi^*}} \left[ \eta \max_{a'} \bar{f}^{(t)}(s,a') \right] + \mathbb{E}_{(s,a) \sim d^{\pi^*}} \left[ \log \pi^{(T+1)}(a \mid s) - \log \pi^{(1)}(a \mid s) \right].$$

Using the fact that $\log(\pi(a \mid s)) \leq 0$, since $\pi(a|s) \leq 1$, and $\pi^{(1)}(a \mid s) = \frac{1}{|\mathcal{A}_{tar}|}$, we have

$$\sum_{t=1}^{T} \mathbb{E}_{(s,a) \sim d^{\pi^*}} \left[ \bar{f}^{(t)}(s,a) \right] \leq \sum_{t=1}^{T} \mathbb{E}_{(s,a) \sim d^{\pi^*}} \left[ \max_{a'} \bar{f}^{(t)}(s,a') \right] + \frac{\log |\mathcal{A}_{tar}|}{\eta}.$$

$$\square$$

**Lemma 3** ((Agarwal et al., 2019), Chapter 4). *Let $\tau = (s_0, a_0, s_1, a_1, \cdots)$ denote the (random) trajectory generated under a policy $\pi$ in an infinite-horizon MDP $\mathcal{M}$. For any function $f : \mathcal{S} \times \mathcal{A} \to \mathbb{R}$, we have*

$$\mathbb{E}_\tau\left[\sum_{t=0}^\infty \gamma^t f(s_t, a_t)\right] = \frac{1}{1-\gamma}\mathbb{E}_{(s,a)\sim d^\pi}\left[f(s,a)\right]. \tag{8}$$

**Lemma 4** (Importance Ratio). *Given a fixed policy $\pi$ and a fixed state-action pair $(s, a)$, let $p_k(s, a)$ denote the probability of reaching $(s, a)$ under an initial distribution $d^\pi$ and policy $\pi$ after $k$ time steps. Then, for any $k \in \mathbb{N}$, we have*

$$\frac{p_k(s, a)}{d^\pi(s, a)} \leq \frac{1}{(1-\gamma)\mu(s, a)}. \tag{9}$$

*Proof.* To begin with, recall the definition of $d^\pi$ as

$$d^\pi(s, a) := (1-\gamma)\Big(\mu(s, a) + \sum_{t=1}^\infty \gamma^t P(s_t = s, a_t = a; \pi)\Big) \equiv \sum_{t=0}^\infty \gamma^t P(s_t = s, a_t = a; \pi). \tag{10}$$

Let $s_{\text{next},k}$ and $a_{\text{next},k}$ denote the state and action after $k$ time steps. Then, we can write down $p_k(s, a)$:

$$p_k(s, a) = \sum_{(s_0, a_0)} \mathbb{P}(s_{\text{next},k} = s, a_{\text{next},k} = a | s_0, a_0; \pi) d^\pi(s_0, a_0) \tag{11}$$

$$= \sum_{(s_0, a_0)} \mathbb{P}(s_{\text{next},k} = s, a_{\text{next},k} = a | s_0, a_0; \pi) \cdot (1-\gamma) \cdot \sum_{t=0}^\infty \gamma^t \mathbb{P}(s_t = s_0, a_t = a_0; \pi) \tag{12}$$

$$= (1-\gamma) \cdot \sum_{t=0}^\infty \gamma^t \sum_{s_0, a_0} \mathbb{P}(s_{\text{next},k} = s, a_{\text{next},k} = a | s_0, a_0; \pi) \cdot \mathbb{P}(s_t = s_0, a_t = a_0; \pi) \tag{13}$$

$$= (1-\gamma) \sum_{t=0}^\infty \gamma^t \mathbb{P}(s_{t+k} = s, a_{t+k} = a; \pi) \tag{14}$$

Then, we have

$$\frac{p_k(s, a)}{d^\pi(s, a)} = \frac{(1-\gamma)\sum_{t=0}^\infty \gamma^t \mathbb{P}(s_{t+k} = s; a_{t+k} = a; \pi)}{(1-\gamma)\sum_{t=0}^\infty \gamma^t \mathbb{P}(s_t = s, a_t = a; \pi)} \tag{15}$$

$$= \frac{\sum_{t=0}^\infty \gamma^t \mathbb{P}(s_{t+k} = s, a_{t+k} = a; \pi)}{\sum_{t=0}^\infty \gamma^t \mathbb{P}(s_t = s, a_t = a; \pi)} \tag{16}$$

$$\leq \frac{\sum_{t=0}^\infty \gamma^t}{\sum_{t=0}^\infty \gamma^t \mathbb{P}(s_t = s; \pi)} \tag{17}$$

$$= \frac{1}{1-\gamma} \cdot \frac{1}{\sum_{t=0}^\infty \gamma^t \mathbb{P}(s_t = s; \pi)} \tag{18}$$

where (17) holds by $\mathbb{P}(s_{t+k} = s, a_{t+k} = a; \pi) \leq 1$ and (18) holds by taking the sum of an infinite geometric sequence. By $\sum_{t=0}^\infty \gamma^t \mathbb{P}(s_t = s, a_t = a; \pi) = \mu_{\text{tar}}(s) + \sum_{t=1}^\infty \gamma^t \mathbb{P}(s_t = s, a_t = a; \pi)$, we have

$$\frac{1}{1-\gamma} \cdot \frac{1}{\sum_{t=0}^\infty \gamma^t \mathbb{P}(s_t = s, a_t = a; \pi)} = \frac{1}{1-\gamma} \cdot \frac{1}{\mu(s, a) + \sum_{t=1}^\infty \gamma^t \mathbb{P}(s_t = s, a_t = a; \pi)} \tag{19}$$

$$\leq \frac{1}{(1-\gamma)\mu(s, a)} \tag{20}$$

where (20) holds by $\sum_{t=1}^\infty \gamma^t \mathbb{P}(s_t = s, a_t = a; \pi) \geq 0$. $\qquad\square$

## B PROOF OF THEOREM 1

Recall that for any policy $\pi$, we use $d^\pi$ to denote the discounted state-action visitation distribution under policy $\pi$ in the target domain.

**Lemma 5.** *Under Algorithm 1, for any $t \in \mathbb{N}$, we have*

$$
\mathbb{E}_{(s,a)\sim d^{\pi^{(t)}}} \left[ \left( \bar{f}^t(s,a) - A^{\pi^t}(s,a) \right)^2 \right]
$$

$$
\leq \frac{4}{(1-\gamma)^2 \mu_{tar,\,min}^2} \mathbb{E}_{(s,a)\sim d^{\pi^{(t)}}} \left[ \left( (1-\alpha(t))\epsilon_{td}^{(t)}(s,a) + \alpha(t)\epsilon_{src,be}^{(t)}(s,a;Q_{src}) \right)^2 \right] \tag{21}
$$

*Proof.* Recall the definitions that $\bar{f}^{(t)}(s,a) := f^{(t)}(s,a) - f^{(t)}(s,\pi^{(t)}(s))$ and $A^{\pi^{(t)}}(s,a) := Q^{\pi^{(t)}}(s,a) - Q^{\pi^{(t)}}(s,\pi^{(t)}(s))$. Then, we have

$$
\mathbb{E}_{(s,a)\sim d^{\pi^{(t)}}} \left[ \left( \bar{f}^{(t)}(s,a) - A^{\pi^{(t)}}(s,a) \right)^2 \right] \tag{22}
$$

$$
= \mathbb{E}_{(s,a)\sim d^{\pi^{(t)}}} \left[ \left( f^{(t)}(s,a) - f^{(t)}(s,\pi^{(t)}(s)) - Q^{\pi^{(t)}}(s,a) + Q^{\pi^{(t)}}(s,\pi^{(t)}(s)) \right)^2 \right] \tag{23}
$$

$$
\leq \mathbb{E}_{(s,a)\sim d^{\pi^{(t)}}} \left[ 2\left( f^{(t)}(s,a) - Q^{\pi^{(t)}}(s,a) \right)^2 + 2\left( Q^{\pi^{(t)}}(s,\pi^{(t)}(s)) - f^{(t)}(s,\pi^{(t)}(s)) \right)^2 \right] \tag{24}
$$

where (24) holds by the fact that $(x+y)^2 \leq 2x^2 + 2y^2$ for any $x,y \in \mathbb{R}$. Then, by linearity of expectation, we obtain

$$
\mathbb{E}_{(s,a)\sim d^{\pi^{(t)}}} \left[ 2\left( f^{(t)}(s,a) - Q^{\pi^{(t)}}(s,a) \right)^2 + 2\left( Q^{\pi^{(t)}}(s,\pi^{(t)}(s)) - f^{(t)}(s,\pi^{(t)}(s)) \right)^2 \right] \tag{25}
$$

$$
= \mathbb{E}_{(s,a)\sim d^{\pi^{(t)}}} \left[ 2\left( f^{(t)}(s,a) - Q^{\pi^{(t)}}(s,a) \right)^2 \right] + \mathbb{E}_{s\sim d^{\pi^{(t)}}} \left[ 2\left( Q^{\pi^{(t)}}(s,\pi^{(t)}(s)) - f^{(t)}(s,\pi^{(t)}(s)) \right)^2 \right] \tag{26}
$$

$$
= \mathbb{E}_{(s,a)\sim d^{\pi^{(t)}}} \left[ 2\left( f^{(t)}(s,a) - Q^{\pi^{(t)}}(s,a) \right)^2 \right] + \mathbb{E}_{s\sim d^{\pi^{(t)}}} \left[ 2 \left[ \mathbb{E}_{a'\sim\pi^{(t)}(s)} \left[ Q^{\pi^{(t)}}(s,a') - f^{(t)}(s,a') \right] \right]^2 \right] \tag{27}
$$

$$
\leq \mathbb{E}_{(s,a)\sim d^{\pi^{(t)}}} \left[ 2\left( f^{(t)}(s,a) - Q^{\pi^{(t)}}(s,a) \right)^2 \right] + \mathbb{E}_{(s,a')\sim d^{\pi^{(t)}}} \left[ 2\left( Q^{\pi^{(t)}}(s,a') - f^{(t)}(s,a') \right)^2 \right] \tag{28}
$$

$$
\leq 4\mathbb{E}_{(s,a)\sim d^{\pi^{(t)}}} \left[ \left( f^{(t)}(s,a) - Q^{\pi^{(t)}}(s,a) \right)^2 \right] \tag{29}
$$

where (28) holds by Jensen's inequality. Then, we proceed to derive an upper bound on $\mathbb{E}_{(s,a)\sim d^{\pi^{(t)}}} \left[ \left( f^{(t)}(s,a) - Q^{\pi^{(t)}}(s,a) \right)^2 \right]$. By the definition of $f^{(t)} := (1-\alpha(t))Q_{tar}^{(t)}(s,a) + \alpha(t)Q_{src}(\phi^{(t)}(s),\psi^{(t)}(a))$, we have

$$
\mathbb{E}_{(s,a)\sim d^{\pi^{(t)}}} \left[ \left( f^{(t)}(s,a) - Q^{\pi^{(t)}}(s,a) \right)^2 \right] \tag{30}
$$

$$
= \mathbb{E}_{(s,a)\sim d^{\pi^{(t)}}} \left[ \left( (1-\alpha(t))Q_{tar}^{(t)}(s,a) + \alpha(t)Q_{src}(\phi^{(t)}(s),\psi^{(t)}(a)) - Q^{\pi^{(t)}}(s,a) \right)^2 \right] \tag{31}
$$

$$
= \mathbb{E}_{(s,a)\sim d^{\pi^{(t)}}} \Big[ \Big( (1-\alpha(t))\big( Q_{tar}^{(t)}(s,a) - r_{tar}(s,a) + r_{tar}(s,a) \big)
$$
$$
+ \alpha(t)\big( Q_{src}(\phi^{(t)}(s),\psi^{(t)}(a)) - r_{tar}(s,a) + r_{tar}(s,a) \big) - Q^{\pi^{(t)}}(s,a) \Big)^2 \Big] \tag{32}
$$

$$
= \mathbb{E}_{(s,a)\sim d^{\pi^{(t)}}} \Big[ \Big( (1-\alpha(t))\big( Q_{tar}^{(t)}(s,a) - r_{tar}(s,a) + r_{tar}(s,a) - \gamma\mathbb{E}_{\substack{s'\sim P_{tar}(\cdot|s,a)\\ a'\sim\pi^{(t)}(\cdot|s')}}[Q_{tar}^{(t)}(s',a')]
$$
$$
+ \gamma\mathbb{E}_{\substack{s'\sim P_{tar}(\cdot|s,a)\\ a'\sim\pi^{(t)}(\cdot|s')}}[Q_{tar}^{(t)}(s',a')] \big) + \alpha(t)\Big( Q_{src}(\phi^{(t)}(s),\psi^{(t)}(a)) - r_{tar}(s,a) + r_{tar}(s,a)
$$
$$
- \gamma\mathbb{E}_{\substack{s'\sim P_{tar}(\cdot|s,a)\\ a'\sim\pi^{(t)}(\cdot|s')}}[Q_{src}(\phi^{(t)}(s'),\psi^{(t)}(a'))] + \gamma\mathbb{E}_{\substack{s'\sim P_{tar}(\cdot|s,a)\\ a'\sim\pi^{(t)}(\cdot|s')}}[Q_{src}(\phi^{(t)}(s'),\psi^{(t)}(a'))] \Big)
$$
$$
- Q^{\pi^{(t)}}(s,a) \Big)^2 \Big]
$$

$$
\tag{33}
$$

$$
\begin{aligned}
= \mathbb{E}_{(s,a)\sim d^{\pi^{(t)}}} \Bigg[ &\Big( \big(1-\alpha(t)\big)\big(Q^{(t)}_{\mathrm{tar}}(s,a) - r_{\mathrm{tar}}(s,a) - \gamma \mathbb{E}_{\substack{s'\sim P_{\mathrm{tar}}(\cdot|s,a)\\ a'\sim\pi^{(t)}(\cdot|s')}}[Q^{(t)}_{\mathrm{tar}}(s',a')]\big) \\
&+ \alpha(t)\Big(Q_{\mathrm{src}}(\phi^{(t)}(s),\psi^{(t)}(a)) - r_{\mathrm{tar}}(s,a) - \gamma\mathbb{E}_{\substack{s'\sim P_{\mathrm{tar}}(\cdot|s,a)\\ a'\sim\pi^{(t)}(\cdot|s')}}[Q_{\mathrm{src}}(\phi^{(t)}(s'),\psi^{(t)}(a'))]\Big) \\
&+ \big(1-\alpha(t)\big)\gamma\mathbb{E}_{\substack{s'\sim P_{\mathrm{tar}}(\cdot|s,a)\\ a'\sim\pi^{(t)}(\cdot|s')}}[Q^{(t)}_{\mathrm{tar}}(s',a')] + \alpha(t)\gamma\mathbb{E}_{\substack{s'\sim P_{\mathrm{tar}}(\cdot|s,a)\\ a'\sim\pi^{(t)}(\cdot|s')}}[Q_{\mathrm{src}}(\phi^{(t)}(s'),\psi^{(t)}(a'))] \\
&+ r_{\mathrm{tar}}(s,a) - Q^{\pi^{(t)}}(s,a)\Big)^2 \Bigg]
\end{aligned}
\tag{34}
$$

$$
\begin{aligned}
= \mathbb{E}_{(s,a)\sim d^{\pi^{(t)}}} \Bigg[ &\Big( \big(1-\alpha(t)\big)\big(Q^{(t)}_{\mathrm{tar}}(s,a) - r_{\mathrm{tar}}(s,a) - \gamma \mathbb{E}_{\substack{s'\sim P_{\mathrm{tar}}(\cdot|s,a)\\ a'\sim\pi^{(t)}(\cdot|s')}}[Q^{(t)}_{\mathrm{tar}}(s',a')]\big) \\
&+ \alpha(t)\Big(Q_{\mathrm{src}}(\phi^{(t)}(s),\psi^{(t)}(a)) - r_{\mathrm{tar}}(s,a) - \gamma\mathbb{E}_{\substack{s'\sim P_{\mathrm{tar}}(\cdot|s,a)\\ a'\sim\pi^{(t)}(\cdot|s')}}[Q_{\mathrm{src}}(\phi^{(t)}(s'),\psi^{(t)}(a'))]\Big) \\
&+ \gamma\mathbb{E}_{\substack{s'\sim P_{\mathrm{tar}}(\cdot|s,a)\\ a'\sim\pi^{(t)}(\cdot|s')}}[f^{(t)}(s',a')] + r_{\mathrm{tar}}(s,a) - Q^{\pi^{(t)}}(s,a)\Big)^2 \Bigg]
\end{aligned}
\tag{35}
$$

where we obtain (32) by adding the dummy terms $\big(1-\alpha(t)\big)\big(-r_{\mathrm{tar}}(s,a)+r_{\mathrm{tar}}(s,a)\big)$ and $\alpha(t)\big(-r_{\mathrm{tar}}(s,a)+r_{\mathrm{tar}}(s,a)\big)$ to the inner part of (31), (33) is obtained by adding $\big(1-\alpha(t)\big)\big(-\gamma\mathbb{E}_{\substack{s'\sim P_{\mathrm{tar}}(\cdot|s,a)\\ a'\sim\pi^{(t)}(\cdot|s')}}[Q^{(t)}_{\mathrm{tar}}(s',a')] + \gamma\mathbb{E}_{\substack{s'\sim P_{\mathrm{tar}}(\cdot|s,a)\\ a'\sim\pi^{(t)}(\cdot|s')}}[Q^{(t)}_{\mathrm{tar}}(s',a')]\big)$ and $\alpha(t)\big(-\gamma\mathbb{E}_{\substack{s'\sim P_{\mathrm{tar}}(\cdot|s,a)\\ a'\sim\pi^{(t)}(\cdot|s')}}[Q_{\mathrm{src}}(\phi^{(t)}(s'),\psi^{(t)}(a'))] + \gamma\mathbb{E}_{\substack{s'\sim P_{\mathrm{tar}}(\cdot|s,a)\\ a'\sim\pi^{(t)}(\cdot|s')}}[Q_{\mathrm{src}}(\phi^{(t)}(s'),\psi^{(t)}(a'))]\big)$ to the inner part of (32), (34) holds by rearranging the terms in (33), and (35) holds by the definition of $f^{(t)}$. Then, by adding $\gamma\mathbb{E}_{\substack{s''\sim P_{\mathrm{tar}}(\cdot|s,a)\\ a''\sim\pi^{(t)}(\cdot|s'')}}[Q^{\pi^{(t)}}(s'',a'')] - \gamma\mathbb{E}_{\substack{s''\sim P_{\mathrm{tar}}(\cdot|s,a)\\ a''\sim\pi^{(t)}(\cdot|s'')}}[Q^{\pi^{(t)}}(s'',a'')]$ to the inner part of (35), we can rewrite (35) as

$$
\begin{aligned}
\mathbb{E}_{(s,a)\sim d^{\pi^{(t)}}} \Bigg[ &\Big( \big(1-\alpha(t)\big)\big(Q^{(t)}_{\mathrm{tar}}(s,a) - r_{\mathrm{tar}}(s,a) - \gamma \mathbb{E}_{\substack{s'\sim P_{\mathrm{tar}}(\cdot|s,a)\\ a'\sim\pi^{(t)}(\cdot|s')}}[Q^{(t)}_{\mathrm{tar}}(s',a')]\big) \\
&+ \alpha(t)\Big(Q_{\mathrm{src}}(\phi^{(t)}(s),\psi^{(t)}(a)) - r_{\mathrm{tar}}(s,a) - \gamma\mathbb{E}_{\substack{s'\sim P_{\mathrm{tar}}(\cdot|s,a)\\ a'\sim\pi^{(t)}(\cdot|s')}}[Q_{\mathrm{src}}(\phi^{(t)}(s'),\psi^{(t)}(a'))]\Big) \\
&+ \gamma\mathbb{E}_{\substack{s'\sim P_{\mathrm{tar}}(\cdot|s,a)\\ a'\sim\pi^{(t)}(\cdot|s')}}[f^{(t)}(s',a')] + r_{\mathrm{tar}}(s,a) - Q^{\pi^{(t)}}(s,a) \\
&+ \gamma\mathbb{E}_{\substack{s''\sim P_{\mathrm{tar}}(\cdot|s,a)\\ a''\sim\pi^{(t)}(\cdot|s'')}}[Q^{\pi^{(t)}}(s'',a'')] - \gamma\mathbb{E}_{\substack{s''\sim P_{\mathrm{tar}}(\cdot|s,a)\\ a''\sim\pi^{(t)}(\cdot|s'')}}[Q^{\pi^{(t)}}(s'',a'')]\Big)^2 \Bigg]
\end{aligned}
\tag{36}
$$

$$
\begin{aligned}
= \mathbb{E}_{(s,a)\sim d^{\pi^{(t)}}} \Bigg[ \Big| &\big(1-\alpha(t)\big)\big(Q^{(t)}_{\mathrm{tar}}(s,a) - r_{\mathrm{tar}}(s,a) - \gamma \mathbb{E}_{\substack{s'\sim P_{\mathrm{tar}}(\cdot|s,a)\\ a'\sim\pi^{(t)}(\cdot|s')}}[Q^{(t)}_{\mathrm{tar}}(s',a')]\big) \\
&+ \alpha(t)\Big(Q_{\mathrm{src}}(\phi^{(t)}(s),\psi^{(t)}(a)) - r_{\mathrm{tar}}(s,a) - \gamma\mathbb{E}_{\substack{s'\sim P_{\mathrm{tar}}(\cdot|s,a)\\ a'\sim\pi^{(t)}(\cdot|s')}}[Q_{\mathrm{src}}(\phi^{(t)}(s'),\psi^{(t)}(a'))]\Big) \\
&+ \gamma\mathbb{E}_{\substack{s'\sim P_{\mathrm{tar}}(\cdot|s,a)\\ a'\sim\pi^{(t)}(\cdot|s')}}[f^{(t)}(s',a')] + r_{\mathrm{tar}}(s,a) - Q^{\pi^{(t)}}(s,a) \\
&+ \gamma\mathbb{E}_{\substack{s''\sim P_{\mathrm{tar}}(\cdot|s,a)\\ a''\sim\pi^{(t)}(\cdot|s'')}}[Q^{\pi^{(t)}}(s'',a'')] - \gamma\mathbb{E}_{\substack{s''\sim P_{\mathrm{tar}}(\cdot|s,a)\\ a''\sim\pi^{(t)}(\cdot|s'')}}[Q^{\pi^{(t)}}(s'',a'')]\Big|^2 \Bigg]
\end{aligned}
\tag{37}
$$

$$\leq \mathbb{E}_{(s,a)\sim d^{\pi^{(t)}}} \Bigg[ \Bigg( \Big| (1-\alpha(t)) \Big( Q_{\text{tar}}^{(t)}(s,a) - r_{\text{tar}}(s,a) - \gamma \mathbb{E}_{\substack{s'\sim P_{\text{tar}}(\cdot|s,a) \\ a'\sim\pi^{(t)}(\cdot|s')}}[Q_{\text{tar}}^{(t)}(s',a')] \Big) \Big|$$

$$+ \Big| \alpha(t) \Big( Q_{\text{src}}(\phi^{(t)}(s), \psi^{(t)}(a)) - r_{\text{tar}}(s,a) - \gamma \mathbb{E}_{\substack{s'\sim P_{\text{tar}}(\cdot|s,a) \\ a'\sim\pi^{(t)}(\cdot|s')}}[Q_{\text{src}}(\phi^{(t)}(s'), \psi^{(t)}(a'))] \Big) \Big|$$

$$+ \Big| \gamma \mathbb{E}_{\substack{s'\sim P_{\text{tar}}(\cdot|s,a) \\ a'\sim\pi^{(t)}(\cdot|s')}}[f^{(t)}(s',a')] + r_{\text{tar}}(s,a) - Q^{\pi^{(t)}}(s,a) \Big| \tag{38}$$

$$+ \Big| \gamma \mathbb{E}_{\substack{s''\sim P_{\text{tar}}(\cdot|s,a) \\ a''\sim\pi^{(t)}(\cdot|s'')}}[Q^{\pi^{(t)}}(s'',a'')] - \gamma \mathbb{E}_{\substack{s''\sim P_{\text{tar}}(\cdot|s,a) \\ a''\sim\pi^{(t)}(\cdot|s'')}}[Q^{\pi^{(t)}}(s'',a'')] \Big| \Big)^2 \Bigg]$$

$$\leq \mathbb{E}_{(s,a)\sim d^{\pi^{(t)}}} \Bigg[ \Bigg( (1-\alpha(t)) \underbrace{\Big| Q_{\text{tar}}^{(t)}(s,a) - r_{\text{tar}}(s,a) - \gamma \mathbb{E}_{\substack{s'\sim P_{\text{tar}}(\cdot|s,a) \\ a'\sim\pi^{(t)}(\cdot|s')}}[Q_{\text{tar}}^{(t)}(s',a')] \Big|}_{=:\epsilon_{\text{td}}^{(t)}(s,a)}$$

$$+ \alpha(t) \underbrace{\Big| \Big( Q_{\text{src}}(\phi^{(t)}(s), \psi^{(t)}(a)) - r_{\text{tar}}(s,a) - \gamma \mathbb{E}_{\substack{s'\sim P_{\text{tar}}(\cdot|s,a) \\ a'\sim\pi^{(t)}(\cdot|s')}}[Q_{\text{src}}(\phi^{(t)}(s'), \psi^{(t)}(a'))] \Big) \Big|}_{=:\epsilon_{\text{src,be}}^{(t)}(s,a;Q_{\text{src}})} \tag{39}$$

$$+ \Big| \gamma \mathbb{E}_{\substack{s'\sim P_{\text{tar}}(\cdot|s,a) \\ a'\sim\pi^{(t)}(\cdot|s')}}[f^{(t)}(s',a')] - \gamma \mathbb{E}_{\substack{s''\sim P_{\text{tar}}(\cdot|s,a) \\ a''\sim\pi^{(t)}(\cdot|s'')}}[Q^{\pi^{(t)}}(s'',a'')] \Big|$$

$$+ \underbrace{\Big| r_{\text{tar}}(s,a) - Q^{\pi^{(t)}}(s,a) + \gamma \mathbb{E}_{\substack{s''\sim P_{\text{tar}}(\cdot|s,a) \\ a''\sim\pi^{(t)}(\cdot|s'')}}[Q^{\pi^{(t)}}(s'',a'')] \Big|}_{=0} \Big)^2 \Bigg]$$

$$= \mathbb{E}_{(s,a)\sim d^{\pi^{(t)}}} \Bigg[ \Bigg( (1-\alpha(t))\epsilon_{\text{td}}^{(t)}(s,a) + \alpha(t)\epsilon_{\text{src,be}}^{(t)}(s,a;Q_{\text{src}})$$

$$+ \gamma \Big| \mathbb{E}_{\substack{s'\sim P_{\text{tar}}(\cdot|s,a) \\ a'\sim\pi^{(t)}(\cdot|s'')}}\Big[ f^{(t)}(s',a') - Q^{\pi^{(t)}}(s',a') \Big] \Big| \Bigg)^2 \Bigg] \tag{40}$$

$$= \mathbb{E}_{(s,a)\sim d^{\pi^{(t)}}} \Bigg[ \Bigg( (1-\alpha(t))\epsilon_{\text{td}}^{(t)}(s,a) + \alpha(t)\epsilon_{\text{src,be}}^{(t)}(s,a;Q_{\text{src}})$$

$$+ \gamma \mathbb{E}_{\substack{s'\sim P_{\text{tar}}(\cdot|s,a) \\ a'\sim\pi^{(t)}(\cdot|s'')}}\Big[ \big| f^{(t)}(s',a') - Q^{\pi^{(t)}}(s',a') \big| \Big] \Bigg)^2 \Bigg] \tag{41}$$

where (37) holds by the fact that $x^2 = |x|^2$, (38) holds by triangle inequality, (39) by the facts that $0 \leq \alpha(t) \leq 1$ and $0 \leq 1-\alpha(t) \leq 1$, (40) holds by coupling $(s',a')$ and $(s'',a'')$ and applying Bellman expectation equation as well as the definitions that $\epsilon_{\text{td}}^{(t)}(s,a) := \big| Q_{\text{tar}}^{(t)}(s,a) - r_{\text{tar}}(s,a) - \gamma \mathbb{E}_{\substack{s'\sim P_{\text{tar}}(\cdot|s,a) \\ a'\sim\pi^{(t)}(\cdot|s')}}[Q_{\text{tar}}^{(t)}(s',a')] \big|$ and $\epsilon_{\text{src,be}}^{(t)}(s,a;Q_{\text{src}}) := \big| Q_{\text{src}}(\phi^{(t)}(s), \psi^{(t)}(a)) - r_{\text{tar}}(s,a) - \gamma \mathbb{E}_{\substack{s'\sim P_{\text{tar}}(\cdot|s,a) \\ a'\sim\pi^{(t)}(\cdot|s')}}[Q_{\text{src}}(\phi^{(t)}(s'), \psi^{(t)}(a'))] \big|$. By recursively applying the procedure from (30) to (41) to $\big| f^{(t)}(s',a') - Q^{\pi^{(t)}}(s',a') \big|$, we obtain a bound on $\mathbb{E}_{(s,a)\sim d^{\pi^{(t)}}} \Big[ \big( f^{(t)}(s,a) - Q^{\pi^{(t)}}(s,a) \big)^2 \Big]$ as follows:

$$\mathbb{E}_{(s,a)\sim d^{\pi^{(t)}}} \Big[ \big( f^{(t)}(s,a) - Q^{\pi^{(t)}}(s,a) \big)^2 \Big] \tag{42}$$

$$\leq \mathbb{E}_{(s,a)\sim d^{\pi^{(t)}}} \Bigg[ \Bigg( (1-\alpha(t))\epsilon_{\text{td}}^{(t)}(s,a) + \alpha(t)\epsilon_{\text{src,be}}^{(t)}(s,a;Q_{\text{src}})$$

$$+ \gamma \mathbb{E}_{\substack{s'\sim P_{\text{tar}}(\cdot|s,a) \\ a'\sim\pi^{(t)}(\cdot|s')}}\Big[ \big| f^{(t)}(s',a') - Q^{\pi^{(t)}}(s',a') \big| \Big] \Bigg)^2 \Bigg] \tag{43}$$

$$
\leq \mathbb{E}_{(s,a)\sim d^{\pi^{(t)}}}\left[\left(\left(1-\alpha(t)\right)\epsilon_{\text{td}}^{(t)}(s,a)+\alpha(t)\epsilon_{\text{src,be}}^{(t)}(s,a;Q_{\text{src}})\right.\right.
$$

$$
+\gamma\mathbb{E}_{\substack{s'\sim P_{\text{tar}}(\cdot|s,a)\\a'\sim\pi^{(t)}(\cdot|s')}}\left[\left(1-\alpha(t)\right)\epsilon_{\text{td}}^{(t)}(s',a')+\alpha(t)\epsilon_{\text{src,be}}^{(t)}(s',a';Q_{\text{src}})\right.
\tag{44}
$$

$$
\left.\left.\left.+\mathbb{E}_{\substack{s''\sim P_{\text{tar}}(\cdot|s',a')\\a''\sim\pi^{(t)}(\cdot|s'')}}\left[\left|f^{(t)}(s'',a'')-Q^{\pi^{(t)}}(s'',a'')\right|\right]\right]\right)^2\right]
$$

$$
\leq \mathbb{E}_{(s,a)\sim d^{\pi^{(t)}}}\left[\left(\left(1-\alpha(t)\right)\epsilon_{\text{td}}^{(t)}(s,a)+\alpha(t)\epsilon_{\text{src,be}}^{(t)}(s,a;Q_{\text{src}})\right.\right.
$$

$$
+\frac{1}{(1-\gamma)\mu_{\text{tar,min}}}\left(\gamma\left(1-\alpha(t)\right)\epsilon_{\text{td}}^{(t)}(s,a)+\gamma\alpha(t)\epsilon_{\text{src,be}}^{(t)}(s,a;Q_{\text{src}})\right.
\tag{45}
$$

$$
\left.\left.\left.+\gamma^2\left(1-\alpha(t)\right)\epsilon_{\text{td}}^{(t)}(s,a)+\gamma^2\alpha(t)\epsilon_{\text{src,be}}^{(t)}(s,a;Q_{\text{src}})+\cdots\right)\right)^2\right]
$$

$$
\leq \frac{1}{(1-\gamma)^4\mu_{\text{tar,min}}^2}\mathbb{E}_{(s,a)\sim d^{\pi^{(t)}}}\left[\left(\left(1-\alpha(t)\right)\epsilon_{\text{td}}^{(t)}(s,a)+\alpha(t)\epsilon_{\text{src,be}}^{(t)}(s,a;Q_{\text{src}})\right)^2\right]
\tag{46}
$$

where (44) holds by applying the procedure from (30) to (41) to $f^{(t)}(s',a')-Q^{\pi^{(t)}}(s',a')$, (45) holds by applying the procedure from (30) to (41) to all the subsequent time steps and using importance sampling with the importance ratio bound in Lemma 4 and then using the same dummy variables $(s,a)$ for all the subsequent state-action pairs, and (46) holds by taking the sum of an infinite geometric sequence. $\qquad\square$

**Theorem 1.** *(Average Sub-Optimality) Under the QAvatar in Algorithm 1 and Assumption 1, given any fixed learning rate $\eta>0$, the average sub-optimality over $T$ iterations can be upper bounded as*

$$
\frac{1}{T}\sum_{t=1}^{T}\left(V^{\pi^*}(\mu_{tar})-V^{\pi^{(t)}}(\mu_{tar})\right)\leq\underbrace{\frac{1}{(1-\gamma)T}\sum_{t=1}^{T}\mathbb{E}_{(s,a)\sim d^{\pi^*}}\left[\max_{a'}\bar{f}^{(t)}(s,a')\right]+\frac{\log|\mathcal{A}_{tar}|}{(1-\gamma)T\eta}}_{(a)}
$$

$$
+\underbrace{\frac{C_0}{T}\sum_{t=1}^{T}\alpha(t)\|\epsilon_{src,be}^{(t)}(Q_{src})\|_\infty}_{(b)}+\underbrace{\frac{C_0}{T}\sum_{t=1}^{T}(1-\alpha(t))\|\epsilon_{td}^{(t)}\|_\infty}_{(c)},
\tag{4}
$$

*where $C_0:=2\sqrt{C_{\pi^*}}/((1-\gamma)^2\mu_{tar,min})$ and $\bar{f}^{(t)}(s,a):=f^{(t)}(s,a)-f^{(t)}(s,\pi^{(t)}(s))$.*

*Proof.* We start by providing an upper bound on the sub-optimality gap $V^{\pi^*}(\mu_{\text{tar}})-V^{\pi^{(t)}}(\mu_{\text{tar}})$ at each iteration. Recall that $d_{\text{tar}}^\pi$ denotes the discounted state-action visitation distribution of policy $\pi$ in the target domain. Note that

$$
V^{\pi^*}(\mu_{\text{tar}})-V^{\pi^{(t)}}(\mu_{\text{tar}})
\tag{47}
$$

$$
=\frac{1}{1-\gamma}\mathbb{E}_{(s,a)\sim d_{\text{tar}}^{\pi^*}}\left[A^{\pi^{(t)}}(s,a)\right]
\tag{48}
$$

$$
=\frac{1}{1-\gamma}\mathbb{E}_{(s,a)\sim d_{\text{tar}}^{\pi^*}}\left[\bar{f}^{(t)}(s,a)-\bar{f}^{(t)}(s,a)+A^{\pi^{(t)}}(s,a)\right]
\tag{49}
$$

$$
=\frac{1}{1-\gamma}\mathbb{E}_{(s,a)\sim d_{\text{tar}}^{\pi^*}}\left[\bar{f}^{(t)}(s,a)\right]+\frac{1}{1-\gamma}\mathbb{E}_{(s,a)\sim d_{\text{tar}}^{\pi^*}}\left[-\bar{f}^{(t)}(s,a)+A^{\pi^{(t)}}(s,a)\right]
\tag{50}
$$

$$
\leq\frac{1}{1-\gamma}\mathbb{E}_{(s,a)\sim d_{\text{tar}}^{\pi^*}}\left[\bar{f}^{(t)}(s,a)\right]+\frac{1}{1-\gamma}\sqrt{\mathbb{E}_{(s,a)\sim d_{\text{tar}}^{\pi^*}}\left[\left(-\bar{f}^{(t)}(s,a)+A^{\pi^{(t)}}(s,a)\right)^2\right]},
\tag{51}
$$

where (48) holds by the performance difference lemma (cf. Lemma 1), (49) is obtained by adding $\bar{f}^t(s,a)-\bar{f}^t(s,a)$, (50) is obtained by rearranging the terms in (49), and (51) holds by Jensen's

inequality. By the fact that $\|\frac{d^{\pi^*}}{d^{\pi^{(t)}}}\|_\infty \leq C$, we have

$$\frac{1}{1-\gamma}\mathbb{E}_{(s,a)\sim d^{\pi^*}}\left[\bar{f}^{(t)}(s,a)\right] + \frac{1}{1-\gamma}\sqrt{\mathbb{E}_{s,a\sim d^{\pi^*}}\left[\left(-\bar{f}^{(t)}(s,a) + A^{\pi^{(t)}}(s,a)\right)^2\right]} \tag{52}$$

$$\leq \frac{1}{1-\gamma}\mathbb{E}_{(s,a)\sim d^{\pi^*}}\left[\bar{f}^{(t)}(s,a)\right] + \frac{1}{1-\gamma}\sqrt{C\cdot\mathbb{E}_{s,a\sim d^{\pi^{(t)}}}\left[\left(-\bar{f}^{(t)}(s,a) + A^{\pi^{(t)}}(s,a)\right)^2\right]}. \tag{53}$$

Recall the definitions of $\epsilon_{\text{td}}^{(t)}(s,a)$ and $\epsilon_{\text{src, be}}^{(t)}(s,a;Q_{\text{src}})$ as

$$\epsilon_{\text{td}}^{(t)}(s,a) := \left|Q_{\text{tar}}^{(t)}(s,a) - r_{\text{tar}}(s,a) - \gamma\mathbb{E}_{\substack{s'\sim P_{\text{tar}}(\cdot|s,a)\\a'\sim\pi^{(t)}(\cdot|s')}}[Q_{\text{tar}}^{(t)}(s',a')]\right|, \tag{54}$$

$$\epsilon_{\text{src,be}}^{(t)}(s,a;Q_{\text{src}}) := \left|Q_{\text{src}}(\phi^{(t)}(s),\psi^{(t)}(a)) - r_{\text{tar}}(s,a) - \gamma\mathbb{E}_{\substack{s'\sim P_{\text{tar}}(\cdot|s,a)\\a'\sim\pi^{(t)}(\cdot|s')}}[Q_{\text{src}}(\phi^{(t)}(s'),\psi^{(t)}(a'))]\right|. \tag{55}$$

Recall that we also define

$$\|\epsilon_{\text{td}}^{(t)}\|_\infty := \max_{(s,a)\in\mathcal{S}\times\mathcal{A}}\epsilon_{\text{td}}^{(t)}(s,a), \tag{56}$$

$$\|\epsilon_{\text{src, be}}^{(t)}(Q_{\text{src}})\|_\infty := \max_{(s,a)\in\mathcal{S}\times\mathcal{A}}\epsilon_{\text{src, be}}^{(t)}(s,a;Q_{\text{src}}). \tag{57}$$

We are ready to put everything together and establish the cumulative sub-optimality. By taking the summation of (53) over iterations, we have

$$\sum_{t=1}^{T}\mathbb{E}_{s\sim\mu_{\text{tar}}}\left[V^{\pi^*}(s) - V^{\pi^{(t)}}(s)\right] \tag{58}$$

$$\leq \sum_{t=1}^{T}\frac{1}{1-\gamma}\mathbb{E}_{(s,a)\sim d^{\pi^*}}\left[\bar{f}^{(t)}(s,a)\right] + \sum_{t=1}^{T}\frac{1}{1-\gamma}\sqrt{C\mathbb{E}_{s,a\sim d^{\pi^{(t)}}}\left[\left(-\bar{f}^{(t)}(s,a) + A^{\pi^{(t)}}(s,a)\right)^2\right]} \tag{59}$$

$$\leq \sum_{t=1}^{T}\frac{1}{1-\gamma}\mathbb{E}_{(s,a)\sim d^{\pi^*}}\left[\max_{a'}\bar{f}^{(t)}(s,a')\right] + \frac{\log|\mathcal{A}_{\text{tar}}|}{(1-\gamma)\eta}$$

$$+ \frac{2\sqrt{C}}{(1-\gamma)^2\mu_{\text{tar,min}}}\sum_{t=1}^{T}\alpha(t)\|\epsilon_{\text{src, be}}^{(t)}(Q_{\text{src}})\|_\infty + \frac{2\sqrt{C}}{(1-\gamma)^2\mu_{\text{tar,min}}}\sum_{t=1}^{T}(1-\alpha(t))\|\epsilon_{\text{td}}^{(t)}\|_\infty. \tag{60}$$

where (59) follows directly from (53) and (60) holds by Lemma 5 and Lemma 2. $\qquad\square$

## C    ADDITIONAL EXPERIMENTAL RESULTS

### C.1    TOY EXAMPLE TO SHOW THE BENEFIT OF CROSS-DOMAIN BELLMAN-LIKE LOSS.

We consider the 3-by-3 grid navigation problem, as shown in Figure 5. In both domains, there are only two actions: 'going top' and 'going right.' The state of the source domain is described in decimal coordinates, while the state of the target domain is described in binary coordinates. The white squares represent obstacles that cannot be traversed. There are three special states: (i) Start state: The episode always begins at this state. (ii) End state: The episode will only end at this state, and the agent will receive an ending reward of +1. (iii) Treasure state: When the agent first navigates to this state, it will receive

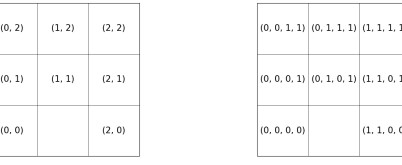

(a) Source Domain          (b) Target Domain

Figure 5: The source and target domain of the grid navigation example.

+0.5 rewards. In other states or at other times navigating the treasure state, the agent will not receive any reward. In the source domain, the start state, end state, and treasure state are set to $(0, 0)$, $(0, 2)$, and $(2, 2)$, respectively. In the target domain, the start state, end state, and treasure state are set to $(0, 0, 0, 0)$, $(0, 0, 1, 1)$, and $(1, 1, 1, 1)$, respectively. We assume that the source Q-function $Q_{src}$ is optimal in the source domain and the environment discount factor $\gamma$ is set to 0.99. It is easy to verify that the optimal trajectory of the source domain is $(0, 0) \rightarrow (0, 1) \rightarrow (0, 2) \rightarrow (1, 2) \rightarrow (2, 2)$ and the optimal trajectory of the target domain is $(0, 0, 0, 0) \rightarrow (0, 0, 0, 1) \rightarrow (0, 0, 1, 1) \rightarrow (0, 1, 1, 1) \rightarrow (1, 1, 1, 1)$. Consider two trajectories in the source domain: Traj-A, which is the optimal trajectory, and Traj-B, defined as $(0, 0) \rightarrow (0, 1) \rightarrow (1, 1) \rightarrow (1, 2) \rightarrow (2, 2)$. When we map the optimal trajectory of the target domain to Traj-A and the optimal trajectory of the target domain to Traj-B, both mappings result in 0 cycle consistency loss. This suggests that the cycle consistency cannot determine which mapping is superior. This phenomenon results from the unsupervised nature of dynamics cycle consistency. In contrast, when we mapping the optimal trajectory of the target domain to Traj-A yields a cross-domain Bellman-like loss of 0, while mapping the optimal trajectory of the target domain to Traj-B results in a cross-domain Bellman-like loss of 1. Thus, we can achieve optimal mapping results based on the cross-domain Bellman error, while the cycle consistency loss provides sub-optimal mapping results.

## C.2 Reproduction and Sanity Checks for DCC, CMD, and CAT

In this section, we report the reproduced scores of DCC, CMD, and CAT as sanity checks for successful reproduction. To simplify the expressions, we abbreviate 'original' as 'orig' and 'reproduced' as 'repr'. As shown in Table 2, we can observe that $Q$Avatar is indeed outperform than DCC, CMD. For the CAT, since the original paper only provides the result of 2-to-1 transfer (i.e., two source domains and one target domain) from CentipedeFour and CentipedeEight to CentipedeSix and get around 2200 episodic return at timesteps 700000. Also, in the original paper, CAT has reproduced the MIKT, which is equivalent to CAT for 1-to-1 transfer (i.e., one source domain and one target domain), and get around 1900 episodic rewards at timesteps 700000. In our reproduction, CAT achieves 1715 evaluation return, and this is close to MIKT in the original CAT paper. Hence, we believe that our implementation of CAT is correct given that our configuration is the same as MIKT.

Table 2: The original and reproduced results of DCC and CMD compared with $Q$Avatar and SAC.

| Environment | DCC (orig) | DCC (repr) | CMD (orig) | CMD (repr) | SAC | $Q$Avatar |
|---|---|---|---|---|---|---|
| Swimmer | $204 \pm 56$ | $132 \pm 47$ | $44 \pm 38$ | $41 \pm 14$ | $235 \pm 141$ | $\mathbf{316 \pm 139}$ |
| Halfcheetah | $2471 \pm 382$ | $1360 \pm 729$ | $2114 \pm 332$ | $303 \pm 75$ | $11445 \pm 1897$ | $\mathbf{12819 \pm 679}$ |
| Ant | N/A | $973 \pm 501$ | $649 \pm 347$ | $882 \pm 52$ | $2290 \pm 785$ | $\mathbf{2840 \pm 1532}$ |

## C.3 Final Rewards

In this section, we show the asymptotic performance of all baselines and our algorithm. In the MuJoCo environments except for Ant and Inverted Double Pendulum, we train all the target-domain models for 500k steps. In Ant and Inverted Double Pendulum, we train all the target-domain models for 350k and 20k steps, respectively. In Robosuite environments, we train all the target-domain models for 20k steps. The asymptotic performances of all baselines and our algorithm are shown in the following tables.

Table 3: Final rewards of $Q$Avatar and all baselines in the MuJoCo environments.

| Algorithm | Hopper | HalfCheetah | Ant | Centipede | Reacher | Modified IDP |
|---|---|---|---|---|---|---|
| $Q$Avatar | $\mathbf{2762 \pm 440}$ | $\mathbf{12316 \pm 586}$ | $\mathbf{2234 \pm 1112}$ | $\mathbf{2020 \pm 1465}$ | $-6.1 \pm 0.3$ | $9241 \pm 62$ |
| SAC | $2086 \pm 257$ | $10986 \pm 1822$ | $1620 \pm 527$ | $872 \pm 36$ | $\mathbf{-5.5 \pm 0.1}$ | $9212 \pm 152$ |
| CMD | $59 \pm 46$ | $-253 \pm 344$ | $778 \pm 144$ | $834 \pm 6116$ | $-14.8 \pm 0.5$ | $72 \pm 12$ |
| DCC | $30 \pm 16$ | $-631 \pm 185$ | $-1240 \pm 838$ | $148 \pm 182$ | $-16.5 \pm 1.4$ | $95 \pm 6$ |
| CAT | $154 \pm 156$ | $46 \pm 250$ | $17 \pm 27$ | $1715 \pm 430$ | $-12.2 \pm 1.0$ | $41 \pm 10$ |
| FT | $2530 \pm 456$ | $12016 \pm 1052$ | $1740 \pm 642$ | $1123 \pm 508$ | $-5.8 \pm 0.2$ | $\mathbf{9349 \pm 86}$ |

Table 4: Final performances of $Q$Avatar and all baselines in the Robosuite and BARK-ML environments.

| Environment | Block Lifting | Door Opening | Table Wiping | Highway |
|---|---|---|---|---|
| QAvatar | **98.0 ± 21.1** | **185.2 ± 66.9** | **67.1 ± 9.1** | **132.2± 31.7** |
| SAC | 90.3 ± 23.4 | 160.1 ± 40.3 | 47.2 ± 7.1 | 117.8 ±15.0 |
| CMD | 0.9 ± 0.6 | 7.8 ± 6.4 | 0.8 ± 0.4 | 13.0 ± 4.7 |
| DCC | 0.6 ± 0.2 | 8.2 ± 4.7 | 0.9 ± 0.7 | 18.1 ± 19.4 |
| CAT | 15.0 ± 14.3 | 34.7 ± 8.4 | 55.5 ± 29.7 | 70.1 ± 6.3 |
| FT | 21.9 ± 7.8 | 129.2 ± 44.9 | 36.8 ± 17.2 | 119.1 ± 21.1 |

## C.4 TIME TO THRESHOLD

In the following table, we discover that $Q$Avatar uses the less data to reach the threshold than SAC does. In Hopper, $Q$Avatar only needs half the amount of data SAC needs to reach the goal.

Table 5: Time to threshold of $Q$Avatar and all baselines

| Environment | Threshold | $Q$Avatar | SAC | SAC / $Q$Avatar |
|---|---|---|---|---|
| Hopper | 2300 | 252K | 836K | 3.32 |
| HalfCheetah | 10000 | 288K | 400K | 1.39 |
| Ant | 1600 | 254K | 344K | 1.35 |
| Centipede | 900 | 210K | 988K | 4.70 |
| Block Lifting | 85 | 90K | 94K | 1.04 |
| Door Opening | 150 | 80K | 94K | 1.18 |
| Table Wiping | 45 | 74K | 96K | 1.30 |
| Highway | 110 | 236K | 374K | 1.58 |
| Reacher | -7 | 150K | 84K | 0.56 |
| Inverted Double Pendulum | 9000 | 16K | 18K | 1.13 |

## C.5 ABLATION STUDY: DEACTIVATING THE FLOW MODEL

As mentioned above, we use a normalizing flow model to restrict the output range of the mapping functions in the feasible regions. In this experiment, we disable the flow model and evaluate $Q$Avatar in Swimmer and Door Opening. In Figure 6, $Q$Avatar without a flow model performs worse than $Q$Avatar with a flow model. In Door Opening, although the ewma values of rewards obtained by $Q$Avatar without the flow model are higher than 100, it has to spend more time attaining high rewards than $Q$Avatar with the flow model does.

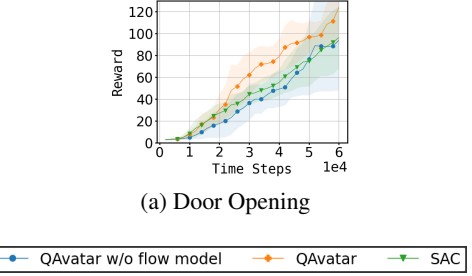

(a) Door Opening

| | | |
|---|---|---|
| QAvatar w/o flow model | QAvatar | SAC |

Figure 6: Ablation Study: $Q$Avatar without the flow model

## D IMPLEMENTATION DETAILS OF $Q$AVATAR

### D.1 PSEUDO CODE OF THE PRACTICAL IMPLEMENTATION OF $Q$AVATAR

In this section, we provide the pseudo code of the practical version of $Q$Avatar in Algorithm 2.

### D.2 INTER-DOMAIN MAPPING NETWORK AUGMENTED WITH A NORMALIZING FLOW MODEL

As mentioned in Section 4, we use the flow model to map the outputs of the mapping functions to the feasible regions. The way to integrate these two components is shown in Figure 7.

---

**Algorithm 2** Practical Implementation of $Q$Avatar

---

**Require:** Source-domain Q-network $Q_{\text{src}}$, value function $V_{\text{src}}$, and the decay function $\alpha : \mathbb{R} \to [0, 1]$.
 1: Initialize the state mapping function $\phi$, the action mapping function $\psi$, the target-domain policy network $\pi$, and entropy coefficient $\beta$
 2: **for** iteration $t = 1, \cdots, T$ **do**
 3:     Sample $\mathcal{D}_{\text{tar}}^{(t)} = \{(s, a, r, s')\}$ of $N_{\text{tar}}$ samples using $\pi^{(t)}$ in the target domain
 4:     Update the target-domain $\{Q_{\text{tar},1}, Q_{\text{tar},2}\}$ by SAC's critic loss:

$$Q_{\text{tar},j}^{(t)} = \arg\min_{Q_{\text{tar}}} \hat{\mathbb{E}}_{(s,a,r,s')\in\mathcal{D}_{\text{tar}}^{(t)}} \left[ \left| r + \gamma\mathbb{E}_{a'\sim\pi^{(t)}} \left[ Q_{\text{tar}}(s', a') - \beta\log(\pi(a'|s')) \right] - Q_{\text{tar}}(s, a) \right|^2 \right]. \tag{61}$$

 5:     Update the state mapping function $\phi$ and action mapping function $\psi$ by minimizing
 6:     the following loss

$$\phi^{(t)}, \psi^{(t)} = \arg\min_{\phi,\psi} \hat{\mathbb{E}}_{(s,a,r,s')\in\mathcal{D}_{\text{tar}}^{(t)}} \left[ \left| r + \gamma V_{\text{src}}(\phi(s')) - Q_{\text{src}}(\phi(s), \psi(a)) \right| \right]. \tag{62}$$

 7:     Update the target-domain policy $\pi$

$$\pi^{(t+1)} = \arg\min_{\pi} \hat{\mathbb{E}}_{(s,a,r,s')\in\mathcal{D}_{\text{tar}}^{(t)}, a'\sim\pi^{(t)}(\cdot|s)} \left[ \beta\log\pi(a'|s) - f^{(t)}(s, a') \right], \tag{63}$$

$$f^{(t)}(s, a') = (1 - \alpha(t)) \min_{j=1,2} Q_{\text{tar},j}^{(t)}(s, a') + \alpha(t)Q_{\text{src}}(\phi^{(t)}(s), \psi^{(t)}(a')). \tag{64}$$

 8: **end for**

---

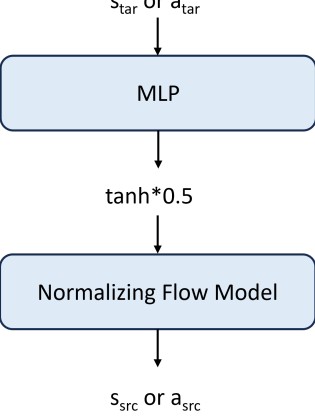

Figure 7: Integration of the mapping function and the normalizing flow model.

# E    CONFIGURATION DETAILS OF THE EXPERIMENTS

## E.1    MUJOCO ENVIRONMENTS

As mentioned in Section 5, the source domains of our experiments are the original MuJoCo environments such as Swimmer-v3, Hopper-v3, HalfCheetah-v3 and Ant-v3. The target domains are the modified MuJoCo environments such as Swimmer with four limbs, Hopper with an extra thigh, HalfCheetah with three legs and Ant with five legs. For the Centipede, CentipedeFour refers to a Centipede with four legs, and CentipedeSix refers to a Centipede with six legs. The environments are shown in Figure 8.

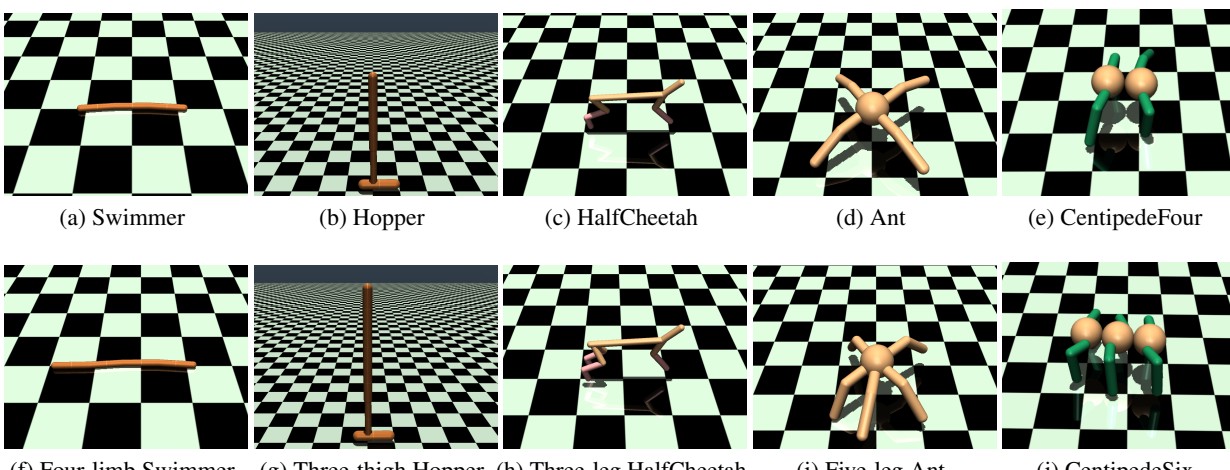

Figure 8: The environments of the source domains and the target domains. (a)-(e): Source domains – Original MuJoCo environments and CentipedeFour. (f)-(j): Target domains – Modified MuJoCo environments and CentipedeSix.

### E.2 ROBOSUITE AND BARK-ML ENVIRONMENTS

Robosuite is a popular robot learning package. We evaluate $Q$Avatar on three tasks, including block lifting, door opening, and table wiping. For each task, we consider cross-domain transfer from controlling a Panda robot arm to controlling a UR5e robot arm. For the BARK-ML environments, we consider transfer from "merging-v0" to "highway-v0". These four tasks are illustrated in Figure 9.

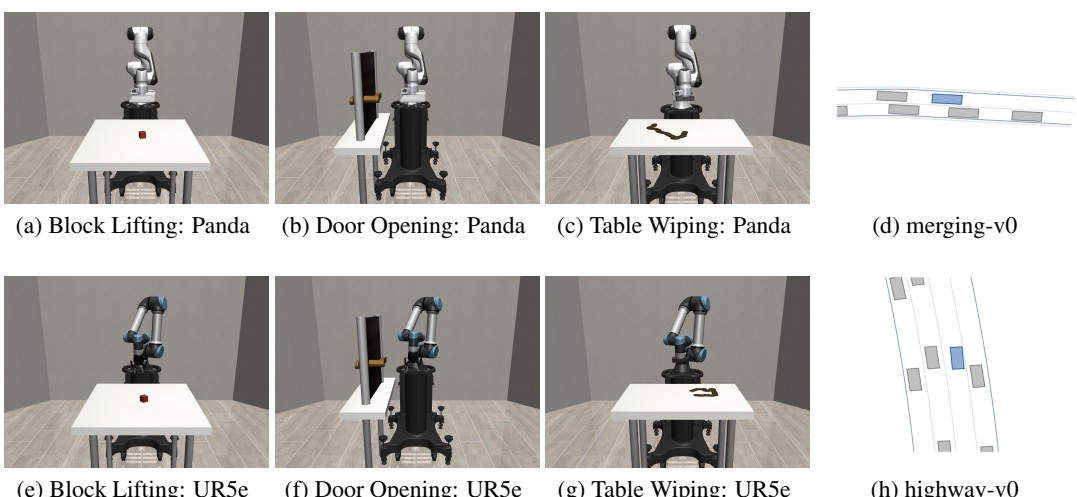

Figure 9: The environments of the source domains and the target domains. (a)-(d)The source domains: control Panda to solve the tasks in robosuite and merging-v0 in bark-ml. (e)-(h)The target domains: control UR5e to solve the tasks in robosuite and highway-v0 in bark-ml.

### E.3 THE IMPLEMENTATION DETAILS OF BASELINES

**SAC.** The implementation of SAC used in our experiments is released by stable-baselines3 Raffin et al. (2021). The settings of all hyperparameters except for the discouted factor $\gamma$ follows the default settings of SAC in the documentation of stable-baselines3. The discouted factor is set as 0.9999

in Swimmer-v3 and 0.99 in all other MuJoCo environments, which follows the setting shown in Hugging Face. As for in the Robosuite environments, we set the discouted factor to 0.9.

**CMD.** We implement CMD by ourselves according to the pseudocode of CMD shown in its original paper Gui et al. (2023). We follow the setting of the hyperparameters which is revealed in its original paper. Additionally, we change CMD from collecting the fixed amount of data to collecting data continuously for a fair comparison. As for the source model, we use the same model used in our algorithm.

**DCC.** We use the original implementation of (Zhang et al., 2021) (`https://github.com/sjtuzq/Cycle_Dynamics`) with their default setting Zhang et al. (2021). For a fair comparison, we use the same source model used in $Q$Avatar and change DCC from collecting the fixed amount of data to collecting data continuously.

**FT.** FT can be seen as a standard SAC algorithm with source feature initialization. Specifically, we modify the input and output layers of the source policy to match the target domain's state and action dimensions, using random initialization, while keeping the middle layers with the same weights as the source model. Similarly, for the source Q function, we adjust the input layer to fit the target domain's state and action dimensions with random initialization, while the remaining layers retain the source model's weights. After initialization, we can use SAC algorithm to implement FT.

**CAT.** We use the authors' implementation (`https://github.com/TJU-DRL-LAB/transfer-and-multi-task-reinforcement-learning/tree/main/Single-agent%20Transfer%20RL/Cross-domain%20Transfer/CAT`) and use PPO as the target-domain base algorithm following the original paper. For a fair comparison, we use the same source model used in $Q$Avatar . The hyperparameters are shown in the following table and "n epochs" means the number of epochs when optimizing the surrogate loss.

Table 6: A list of candidate hyperparameters for Robosuite and MuJoCo.

| Parameter | MuJoCo | Robosuite |
|---|---|---|
| learning rate | 0.0001, 0.0003, 0.0004, 0.0008 | 0.0001, 0.0003 |
| length of rollouts | 500, 2000 (50, 100 for Modified IDP) | 2000 |
| batch size | 50, 100 (20, 25 for Modified IDP) | 50, 100, 200 |
| entropy coefficient (ent. coef.) | 0.01, 0.002 | 0.01, 0.002 |
| n epochs | 10, 20 | 5, 10 |
| num. of hidden layer of encoder/decoder | 1 | 1 |
| num. of hidden layer of actor/critic | 2 | 2 |
| hidden layer size | 256 | 256 |

Table 7: Final hyperparameters chosen for each environment.

| | learning rate | len. of rollouts | batch size | ent. coef. | n epochs |
|---|---|---|---|---|---|
| Hopper | 0.0008 | 2000 | 100 | 0.002 | 20 |
| HalfCheetah | 0.0001 | 500 | 50 | 0.002 | 10 |
| Ant | 0.0004 | 500 | 50 | 0.002 | 10 |
| CentipedeSix | 0.0003 | 2048 | 64 | 0.00 | 10 |
| InvertedDoublePendulum | 0.001 | 100 | 20 | 0.01 | 20 |
| Reacher | 0.0003 | 2048 | 64 | 0.00 | 10 |
| Robosuite | 0.0003 | 2000 | 100 | 0.01 | 10 |
| Highway | 0.0003 | 2048 | 64 | 0.00 | 10 |

## E.4 DETAILED CONFIGURATION OF $Q$AVATAR

The base algorithm, SAC, is implemented by stable-baselines3 Raffin et al. (2021). As for the compute resource, we use NVIDIA GeForce RTX 3090 to do the experiments. Finishing the whole training process including training the source-domain model, target-domain model and flow model once needs 44 hours in the MuJoCo environments and 39 hours in the Robosuite environments. The Hyperparameters of $Q$Avatar are shown in the following two tables. The consider functions of decay functions are $1/\sqrt{t}, 1/t, 1/t^2$ and $1/t^3$ and the final decay functions chosen for each environments are shown in the table 9. The settings of hyperparameters such as critic/actor learning rate, batch size, buffer size and discounted factor are same as SAC.

Table 8: A list of hyperparameters of $Q$Avatar .

| Parameter | Value |
| --- | --- |
| critic/actor learning rate | 0.0003 |
| state mapping function learning rate | 0.01 |
| action mapping function learning rate | 0.01 |
| batch size | 256 |
| replay buffer size | $10^6$ |
| optimizer | Adam |
| number of hidden layer of mapping functions | 1 |
| hidden layer size | 256 |

Table 9: A list of environment-specific hyperparameters of $Q$Avatar .

| Environment | Decay Function $\alpha$ |
| --- | --- |
| Hopper-v3 | $1/t^2$ |
| HalfCheetah-v3 | $1/t^3$ |
| Ant-v3 | $1/\sqrt{t}$ |
| CentipedeSix | $1/t$ |
| InvertedDoublePendulum-v2 | $1/t$ |
| Reacher-v2 | $1/t^3$ |
| Block Lifting | $1/t$ |
| Door Opening | $1/t$ |
| Table Wiping | $1/t$ |
| Highway-v0 | $1/t^3$ |

# F EXPERIMENTS RESULTS FOR REBUTTAL

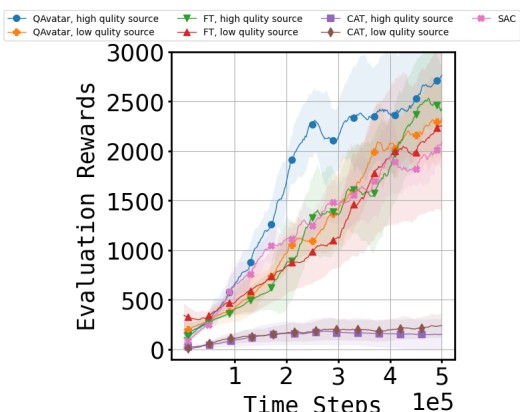

Figure F.1: Training curves of $Q$Avatar and baselines using high/low-quality pre-trained models in Hopper environment.

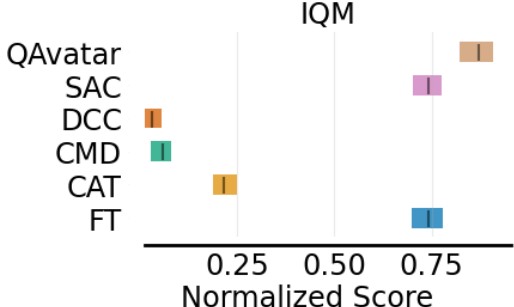

Figure F.2: The aggregated metrics (with 95% stratified bootstrap CIs) for all experiments.

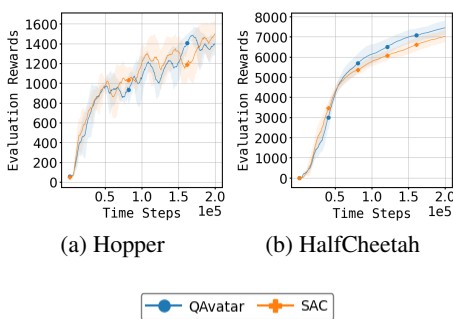 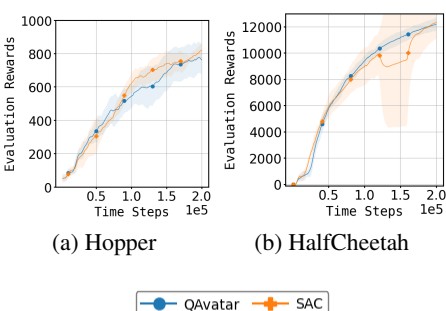

(a) Hopper      (b) HalfCheetah        (a) Hopper      (b) HalfCheetah

Figure F.4: Training curves of QAvatar and SAC in the negative transfer scenario of locomotion tasks.

Figure F.5: Training curves of QAvatar and SAC in the opposite Q-function of source and target domain Transfer Scenario.

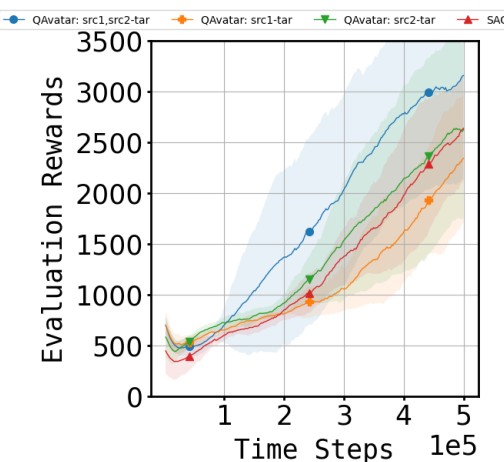

Figure F.3: Training Curves of SAC and $Q$Avatar in three different setting, where "src1-tar" refer to transfer domain src1 to domain tar.(src1: Ant-v3 with front left and back right legs disabled, src2: Ant-v3 with front right and back left Legs disable, tar: original Ant-v3).

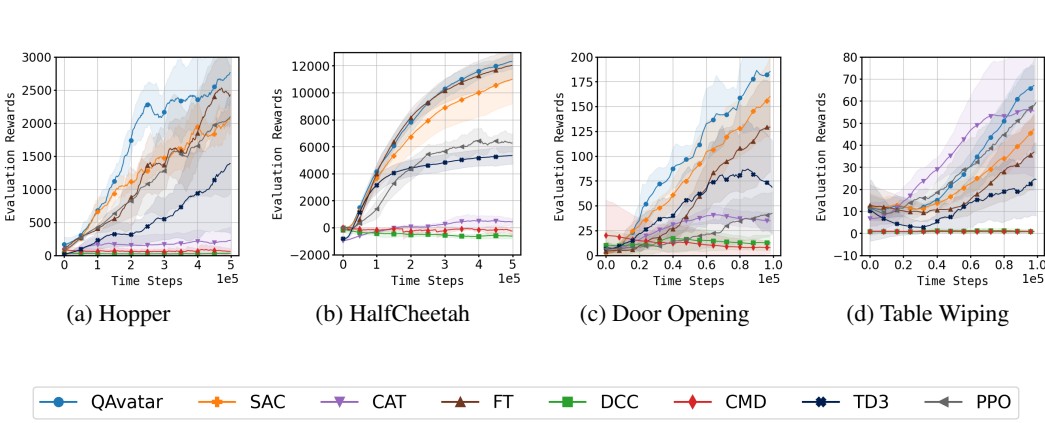

Figure F.6: Training curves of $Q$Avatar and various baseline algorithms are presented. Unlike Figure 1, this version includes two additional baselines: TD3 and PPO, providing a broader comparison of performance across different methods.

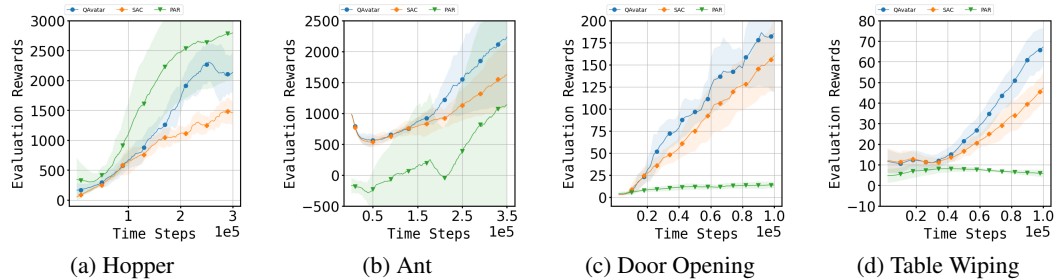

Figure F.7: Training Curves of $Q$Avatar , SAC, and PAR in Various Environments Under the Same Settings as in Section 5.

