# OpenReview forum: "Cross-Domain Reinforcement Learning Under Distinct State-Action Spaces Via Hybrid Q Functions"
_ICLR.cc/2025/Conference — Submitted to ICLR 2025_

### Official Review · Reviewer_qMj1 · 2024-10-26

**Soundness:** 3
**Presentation:** 3
**Contribution:** 2
**Rating:** 6
**Confidence:** 3

**Summary:**

This paper addresses cross-domain transfer in reinforcement learning (RL), particularly when the source and target domains have distinct state and action spaces. The authors propose CDRL, which combines Q-functions from the source and target domains, demonstrating its benefits in various RL tasks, including locomotion, robotic manipulation, and autonomous driving.

**Strengths:**

- The paper is generally well-written, with clear definitions and proofs.
- The authors provide sufficient experiments on diverse tasks, including locomotion, robotic manipulation, and autonomous driving.

**Weaknesses:**

- QAvatar shows a slight improvement in data efficiency (as shown in Figure 1); however, it still requires a similar amount of training time as SAC, which learns from scratch. Since QAvatar involves training in both the source and target domains, it doubles the training time compared to SAC, which may achieve similar performance without requiring as much training time. This makes the claim of improved data efficiency questionable.
- In a similar context to the previous question, it seems that QAvatar requires full training in the source domain. According to Figure 4, if the source-domain model is not fully trained (e.g., of low quality), QAvatar provides no benefits in such cases; instead, it is more practical to train the model directly in the target doamin from scratch. Then, what is the benifict of the cross domain transfer in this context? Could the authors provide additional experiments with other baselines, such as FT or CAT, under the same settings?”
- Can the authors provide experiments on negative transfer in other locomotion tasks, such as Hopper, HalfCheetah, and Ant? Are there specific scenarios where QAvatar may fail to transfer effectively to the target domain?
- Can the authors provide more explanation for why the baseline methods, such as CAT, DCC, and CMD, fail to transfer? It appears that CMD use the same locomotion tasks in their original papers, but no performance improvement is observed in Figure 1.

**Questions:**

- Can the authors provide comparison with other RL algorithm as baseline such as TD3 or PPO?

---

> ### Author Response · Authors · 2024-11-23
> **Rebuttal by Authors**
>
> **Q1: QAvatar shows a slight improvement in data efficiency (as shown in Figure 1); however, it still requires a similar amount of training time as SAC, which learns from scratch. Since QAvatar involves training in both the source and target domains, it doubles the training time compared to SAC, which may achieve similar performance without requiring as much training time. This makes the claim of improved data efficiency questionable.**
>
> A1: Thank you for the thoughtful feedback. In the context of cross-domain transfer for RL, there are two major ways to measure the improvement in data efficiency:
>
> 1. **Asymptotic total reward performance**: To provide a more reliable statistical analysis of asymptotic performance, we follow the general methodology proposed by [Agarwal et al., 2021] and calculate the Interquartile Mean (IQM) using rliable (https://github.com/google-research/rliable). This approach enables better evaluation of results at an aggregated level. Due to the varying reward ranges in different environments, we normalized the total reward to the range [0,1] in each environment to allow comparisons across environments.(https://imgur.com/a/65TrGGM and also in Figure F.2 in Appendix F) shows the aggregated IQMs for all environments, including 95% stratified bootstrap confidence intervals. From the results, we observe that QAvatar achieves significantly better asymptotic performance compared to SAC trained from scratch.
> 2. **Time to thresholds**: This evaluation metric measures the number of environment steps needed to reach a certain level of performance. We summarize this in the table below (also in Section C.4 in Appendix C).
>
> | Environment   | Threshold | QAvatar |   SAC   | QAvatar/SAC  |
> | -----------   | --------- | ------- | ------- |  ----------- |
> | Centipede     |    900    |  210K   |   988K  | 0.21         |
> | Hopper        |   2300    |  252K   |   836K  | 0.30         |
> | Highway       |    110    |  236K   |   374K  | 0.63         |
> | HalfCheetah   |  10000    |  288K   |   400K  | 0.72         |
> | Ant           |   1600    |  254K   |   344K  | 0.74         |
> | Table Wiping  |     45    |   74K   |    96K  | 0.77         |
> | Door Opening  |    150    |   80K   |    94K  | 0.85         |
> | Block Lifting |     85    |   90K   |    94K  | 0.96         |
>
> The above results indicate that in most environments, QAvatar requires fewer target-domain data samples compared to SAC to achieve the target threshold of evaluation reward. Notably, in the Hopper environment, QAvatar requires about only one-third of the target-domain environment steps needed by SAC to reach the same threshold, and in the Centipede environment, it requires even less than one-fourth of the target-domain environment steps.
>
> [Agarwal et al., 2021] R. Agarwal, R., M. Schwarzer, P. S. Castro, A. C. Courville, and M. Bellemare, “Deep reinforcement learning at the edge of the statistical precipice,” NeurIPS 2021.
>
> **Q2: In a similar context to the previous question, it seems that QAvatar requires full training in the source domain. According to Figure 4, if the source-domain model is not fully trained (e.g., of low quality), QAvatar provides no benefits in such cases; instead, it is more practical to train the model directly in the target domain from scratch. Then, what is the benefit of the cross domain transfer in this context? Could the authors provide additional experiments with other baselines, such as FT or CAT, under the same settings?**
>
> A2: Thank you for the insightful question and suggestion.
> - Recall that the main idea of QAvatar is that: (i) If the source-domain model provides positive transfer, QAvatar can effectively leverage the source knowledge (i.e., $Q_{src}$​) to accelerate learning in the target domain. (ii) Otherwise, if the source-domain model introduces negative effects, QAvatar can effectively mitigate these issues and achieve a total reward comparable to SAC through the decay function \alpha(t).
> - Accordingly, the main purpose of Figure 4 is to corroborate that QAvatar remains reliable even in the scenario of negative transfer. Notably, this is an important property since in practice it can be difficult to pre-determine whether the two domains are subject to positive or negative transfer.
> - Moreover, as suggested by the reviewer, to further strengthen the study on low-quality source models, we further evaluate FT and CAT under the same settings in the Hopper environment.
>
> Results are available at https://imgur.com/a/lompKWY (and also in Figure F.1 in Appendix F).
> From these results, we can observe that QAvatar indeed enjoys favorable transfer capability under both high-quality and low-quality source models compared to FT and CAT.

---

> ### Author Response · Authors · 2024-11-23
> **Rebuttal by Authors**
>
> **Q3: Can the authors provide experiments on negative transfer in other locomotion tasks, such as Hopper, HalfCheetah, and Ant? Are there specific scenarios where QAvatar may fail to transfer effectively to the target domain?**
>
> A3:
> Thank you for the helpful suggestion. We provide additional experimental results on the following transfer scenarios
>
> - **Halfcheetah**: The source domain is the original HalfCheetah-v3, and the target domain is HalfCheetah-v3 with the goal of running backward (i.e., adding a negative sign to the forward_reward when calculating the reward). In this scenario, if we deactivate both the inter-domain state and action mappings, it becomes a negative transfer scenario for QAvatar, as the source and target domains have completely opposite goals.
>
> The result is shown in https://imgur.com/a/CPhq8Wi (and also in Figure F.4 of Appendix F).
>
> In the early stages, QAvatar experiences some slight negative effects; however, when $\alpha(t)$ becomes sufficiently small (i.e., for large $t$), QAvatar exhibits a learning curve similar to SAC, despite the negative transfer scenario. This observation aligns with the claims in the key implications of Theorem 1. QAvatar may fail to transfer effectively when negative transfer occurs. During the early stage of the learning process, QAvatar might experience negative effects from the source model. However, as the training time $t$ increases, $\alpha(t)$ decreases and approaches 0. At this point, QAvatar reverts to the original SAC and can achieve the similar convergence behavior.
>
> - **Hopper**: We consider the same setting as in HalfCheetah but replace HalfCheetah-v3 with Hopper-v3, keeping the rest of the setup unchanged. We observe that, in this environment, QAvatar can still achieve a similar convergence behavior to SAC and does not get stuck in a sub-optimal policy, even when the source-domain model introduces a negative effect on the target-domain model.
>
> **Q4: Can the authors provide more explanation for why the baseline methods, such as CAT, DCC, and CMD, fail to transfer? It appears that CMD uses the same locomotion tasks in their original papers, but no performance improvement is observed in Figure 1.**
>
> A4:  As described in Q1 of the Global Response regarding, the main reason for the ineffective transfer of DCC and CMD is that they do not utilize any target reward information during training. This unsupervised problem formulation makes it more challenging for DCC and CMD to learn in the target domain. Regarding CAT, the performance can be attributed to its parameter-based transfer and heuristic nature as well as the slight correlation with its base algorithm.
>
> Notably, for the implementation of CMD, we reproduce CMD by referring to the source code of DCC as they are similar and there is no CMD implementation available. We also follow the same hyperparameter configurations as listed in the CMD papers. With that said, we have tried our best in reproducing the baseline results.
>
> **Q5: Can the authors provide comparison with other RL algorithms as baseline such as TD3 or PPO?**
>
> A5: Thank you for the helpful suggestion. We further evaluate the PPO and TD3 algorithms on two MuJCo tasks (Hopper and HalfCheetah) as well as two Robosuite tasks (Door Opening and Table Wiping) by training in the target domain from scratch (exactly like SAC).
>
> The results are available at https://imgur.com/a/347qFho (and also in Figure F.6 in Appendix F).
>
> We observe that QAvatar still performs consistently better than TD3 and PPO in all these environments. Thus, we can conclude that QAvatar indeed achieves data efficiency compared to the popular benchmark RL algorithms trained from scratch.

---

> > ### Comment · Reviewer_qMj1 · 2024-11-25
> >
> > Thank you for providing a detailed response. My main concerns, particularly regarding Q1 and Q2, have been addressed. As a result, I am raising my score to 6.

---

> ### Author Response · Authors · 2024-12-03
> **Response to Reviewer qMj1**
>
> We thank the reviewer again for all the detailed review and the efforts put into helping us to improve our submission.

---

### Official Review · Reviewer_y8Ga · 2024-11-03

**Soundness:** 3
**Presentation:** 4
**Contribution:** 3
**Rating:** 6
**Confidence:** 3

**Summary:**

This paper proposed a new framework for Cross-Domain Reinforcement Learning (CDRL) named QAvatar, which provides reliable knowledge transfer from a source-domain to a target domain via a weighted combination of the given source-domain Q function $Q_{src}$ and the learned target-domain Q function $Q_{tar}$. The proposed method can handle distinct state and action space between the source and target domains with performance guarantee. Additionally, the paper presents a practical implementation of QAvatar using a normalizing-flow-based state-action mapping and demonstrates its superior performance over a range of baseline methods through extensive experiments across various environments.

**Strengths:**

- The paper is well-written and easy to follow. The notations are well-explained, visualizations are clear and helpful in supporting the conclusions.
- The motivation and main idea of the proposed method is clear and practical.
- The proposed method can improve sample efficiency and avoid negative transfer, the authors provide both theoretical analysis and empirical experiments across a rich set of environments with various state and action dimensions. The conclusion from the empirical experiments are well aligned with the theoretical analysis.
- The proposed method is robust to low quality source-domain policy as well as negative transfer scenarios.
- Authors provide reproduction and sanity checks for their baseline methods.

**Weaknesses:**

- The performance margin between QAvatar and some other baselines various dramatically across different environments. A more thorough explanation will be helpful.
- The proposed method still relies on the assumption that the target-domain agent has access to $Q_{src}$ and $V_{src}$ in addition to $\pi_{src}$. What if we only have access to a pre-trained policy from the source-domain and nothing else (that is more practical in most real-world application scenarios).
- The proposed framework can only handle knowledge transfer from one source-domain and one target-domain, and the inter-domain mapping is required to be trained from scratch for every source-target pair.

**Questions:**

- Why CAT outperformed QAvatar in Table wiping task, while QAvatar has significant advantages over CAT in other two Robosuite environments block lifting and door opening? block stacking and door opening should be harder than table wiping and they have higher state and action dimensions as well.
- What will happen if an adversarial source-domain policy is given ($Q_{src}$ is "opposite" to $Q_{tar}$ in some sense), will it harm the training process via the weighted sum mechanism between the $Q_{src}$ and $Q_{tar}$?

---

> ### Author Response · Authors · 2024-11-23
> **Rebuttal by Authors**
>
> **Q1: The performance margin between QAvatar and some other baselines varies dramatically across different environments. A more thorough explanation will be helpful.**
>
> A1:
> * As described in Q1 of the Global Response, the main reason for the poor performance of DCC and CMD is that they do not utilize any target reward information during training. This unsupervised problem formulation makes it more challenging for DCC and CMD to learn in the target domain. Regarding CAT, the performance can be attributed to its parameter-based transfer and heuristic nature as well as the slight correlation with its base algorithm.
> * **Moreover, as illustrated in Q2 of the Global Response, we show that the overall advantage of QAvatar is indeed statistically significant compared to other baselines.** Specifically, for a more reliable comparison, we follow the methodology proposed by [Agarwal et al., 2021] and calculate the Interquartile Mean (IQM) over all the environments using rliable (https://github.com/google-research/rliable). We observe that QAvatar indeed achieves the best performance at an aggregate level.
>
> [Agarwal et al., 2021] R. Agarwal, R., M. Schwarzer, P. S. Castro, A. C. Courville, and M. Bellemare, “Deep reinforcement learning at the edge of the statistical precipice,” NeurIPS 2021.
>
>
> **Q2: The proposed method still relies on the assumption that the target-domain agent has access to Q_{src} and V_{src} in addition to \pi_{src}.  What if we only have access to a pre-trained policy from the source-domain and nothing else (that is more practical in most real-world application scenarios).**
>
> A2: Thank you for pointing this out. In the cross-domain RL literature, based on the available source-domain knowledge, there are two widely adopted formulations as follows:
> - **Formulation 1 –  Source-domain policy, source-domain value functions (either Q function or state value function), and a source-domain dataset are available:** Prior works built on this formulation include [Song et al., 2016; Heng et al., 2022; Sreenivasan et al., 2023; Yang et al., 2024].
> - **Formulation 2 –  Source-domain policy and a source-domain dataset are available:** Prior works built on this formulation include [Ammar et al., 2015; Zhang et al.,2021; Gui et al., 2023; Zhu et al., 2024].
>
> In the QAvatar framework, only access to the source-domain value functions is required. Therefore, QAvatar is directly applicable under Formulation 1. In Formulation 2 (i.e., we only have a pre-trained policy and a source-domain dataset), one can first perform (off-policy) policy evaluation based on the source-domain data to obtain $Q_{src}$ and $V_{src}$​ for the source-domain policy.
>
> [Song et al., 2016] Jinhua Song, Yang Gao, Hao Wang, and Bo An, “Measuring the distance between finite markov decision processes”, AAMAS 2016.
>
> [Heng et al., 2022]  You Heng, Tianpei Yang, Yan Zheng, Jianye Hao, and Matthew E. Taylor, “Cross-domain adaptive transfer reinforcement learning based on state-action correspondence,” UAI 2022.
>
> [Yang et al., 2024] Tianpei Yang, Heng You, Jianye Hao, Yan Zheng, and Matthew E. Taylor, “A Transfer Approach Using Graph Neural Networks in Deep Reinforcement Learning,” AAAI 2024.
>
> [Sreenivasan et al., 2023]  Ram Ananth Sreenivasan, Hyun-Rok Lee, Yeonjeong Jeong, Jongseong Jang, Dongsub Shim, and Chi Guhn Lee, “A learnable similarity metric for transfer learning with dynamics mismatch,” In PRL Workshop Series–Bridging the Gap Between AI Planning and Reinforcement Learning, 2023.
>
> [Ammar et al., 2015] Haitham Bou Ammar, Eric Eaton, Paul Ruvolo, and Matthew Taylor, “Unsupervised cross-domain transfer in policy gradient reinforcement learning via manifold alignment,” AAAI 2015.
>
> [Zhang et al.,2021] Qiang Zhang, Tete Xiao, Alexei A Efros, Lerrel Pinto, and Xiaolong Wang, “Learning cross-domain correspondence for control with dynamics cycle-consistency,” ICLR 2021
>
> [Gui et al., 2023] Haiyuan Gui, Shanchen Pang, Shihang Yu, Sibo Qiao, Yufeng Qi, Xiao He, Min Wang, and Xue Zhai, “Cross-domain policy adaptation with dynamics alignment. Neural Networks 2023.
>
> [Zhu et al., 2024] Ruiqi Zhu, Tianhong Dai, and Oya Celiktutan, “Cross domain policy transfer with effect cycle-consistency,” ICRA 2024.

---

> ### Author Response · Authors · 2024-11-23
> **Rebuttal by Authors**
>
> **Q3: The proposed framework can only handle knowledge transfer from one source domain and one target domain.**
>
> A3: Thanks for the question. In Section 6 (i.e., Concluding Remarks and Limitations) of the paper , we mentioned that in this paper, QAvatar currently considers only one source domain and one target domain. Despite this, we remark that it is directly feasible to extend the QAvatar framework to multiple source domains by training the target policy using a weighted sum of multiple $Q_{src}$​ from different source domains and $Q_{tar}$ from target domain​. More specifically, in each iteration t, we can update the policy by
> $$
> \pi^{(t+1)}(a \mid s) \propto \pi^{(t)}(a \mid s) \exp \left( \eta \cdot \left( (1 - \alpha^{(t)}) Q_{\text{tar}}^{(t)}(s, a) + \alpha^{(t)} Q_{\text{src}}^{(t)}(s,a) \right) \right)
> $$
> ,where $ \alpha: \mathbb{N} \rightarrow \left[0,1\right]$ is the weight decay function between the $Q_{tar}^{(t)}$ and  $Q_{\text{src}}^{(t)}$, $Q_{\text{src}}^{(t)}(s,a) = w_1^{(t)} Q_{\text{src,1}}(\phi_1^{(t)} (s), \psi_1^{(t)} (a)) + \cdots + w_N^{(t)} Q_{\text{src,N}} (\phi_N^{(t)} (s), \psi_N^{(t)} (a))$, $w_1^{(t)},  \cdots , w_N^{(t)}$ is the weight between all source Q functions constrained by $w_1^{(t)} +\cdots + w_{N}^{(t)} = 1$ and $Q_{src,i}, \phi_{i}^{(t)}, \psi_{i}^{(t)} $ are the $i$-th source-domain Q function, state mapping function and action mapping function at time step $t$, respectively.
>
> To verify this, we also evaluate QAvatar on a two-source to one-target transfer scenario:
> - Source domain 1 (referred to as “src1” in the figures below) is Ant-v3 with the front left and back right legs disabled
> - Source domain 2 (referred to as ‘src2’ in the figures below) is Ant-v3 with the front right and back left legs disabled.
> - Target domain (referred to as ‘tar’ in the figures below) is the original Ant-v3 with no modifications.
>
> Intuitively, combining the knowledge from both src1 and src2 could achieve better transfer to the target domain. As a preliminary study, we simply assign equal weights to $Q_{src,1}$​ and $Q_{src,2}$ (i.e., $w^{(t)}_1=0.5, w^{(t)}_2=0.5$, for all $t$), without any hyperparameter tuning.
> Results are available at https://imgur.com/a/6X2MVfW (also in Figure F.3 in Appendix F of the updated manuscript).
>
> The above results demonstrate that QAvatar can indeed be readily extended to multiple source domains and thereby achieve even better transfer performance.
> On top of this, we remark that assigning equal weights to $Q_{src,i}$ may not be the optimal choice; the underlying weights among the source Q-functions can possibly be further optimized. Based on this, the QAvatar framework offers a promising research direction for addressing cross-domain RL transfer for multiple source and target domains.
>
> **Q4: Why CAT outperformed QAvatar in Table wiping task, while QAvatar has significant advantages over CAT in other two Robosuite environments block lifting and door opening? Block stacking and door opening should be harder than table wiping and they have higher state and action dimensions as well.**
>
> A4:
> As described in Q1 of the General Response, the large variation in CAT’s transfer performance across environments can be mainly attributed to its parameter-based transfer and its heuristic nature, especially when the domains are fairly dissimilar. We use the learning curves for four environments (i.e., HalfCheetah, Hopper, Door Opening, and Table Wiping) to further describe this:
>
> Results are available at https://imgur.com/a/347qFho (and also in Figure F.6 in Appendix F).
>
> * **Table Wiping:** We can observe that both CAT and PPO perform quite well. As CAT is a PPO-based algorithm, we conjecture that CAT could partially benefit from the inherent strength of PPO on the task of Table Wiping.
> * **Hopper and Halfcheetah:** While PPO performs reasonably well, CAT does not achieve effective transfer. This can be attributed to the parameter-based transfer and its heuristic nature of CAT.
> * **Door Opening:** We observe that both CAT and PPO perform poorly. This could be the consequence of both the design of CAT as well as the inherent weakness of PPO on this task.

---

> ### Author Response · Authors · 2024-11-23
> **Rebuttal by Authors**
>
> **Q5: What will happen if an adversarial source-domain policy is given (Q_{src}  is "opposite" to Q_{tar} in some sense), will it harm the training process via the weighted sum mechanism between the Q_{src} and Q_{tar}?**
>
> A5: Thanks for the insightful question. To provide a more comprehensive analysis of this, we consider two scenarios where $Q_{src}$ and $Q_{tar}$ have opposite behavior in some sense:
>
> - In the negative transfer experiment (cf. Figure 2 in the original manuscript), we used Inverted Pendulum (IP) as the source domain and Inverted Double Pendulum (IDP) as the target domain. In this experiment, we modified the action semantics of IDP so that it is physically opposite to IP. Specifically, in IP, the action range is [−1,1]: negative values represent applying force to the cart to the left, while positive values represent applying force to the right. In IDP, we reversed this: negative values represent applying force to the cart to the right, and positive values represent applying force to the left. We observed that when the action decoder cannot be learned, $Q_{src}$ is "opposite" to $Q_{tar}$. Nevertheless, as shown in Figure 2 of the original manuscript, QAvatar can still learn effectively and achieve a total reward comparable to SAC. A similar observation can be made in the Reacher environment (cf. Figure 2(a)).
> - Additionally, we run QAvatar in two environments. The first is HalfCheetah, where the source domain is the original HalfCheetah-v3, and the target domain is a three-legged HalfCheetah with the goal of running backward (i.e., adding a negative sign to the forward reward in the reward function). The second is Hopper, where the source domain is the original Hopper-v3, and the target domain is a three-length-tall Hopper with the goal of running backward. In both scenarios, $Q_{src}$​ is also opposite to $Q_{tar}$​ in the sense that their objectives are entirely contradictory.
> The results are available at https://imgur.com/a/i21UPtr (and also in Figure F.5 Appendix F).
> We observe that the QAvatar’s learning process is not negatively affected by $Q_{src}$, even though there may not be an inherent positive transfer in this case.

---

> > ### Comment · Reviewer_y8Ga · 2024-11-25
> > **Response to Authors**
> >
> > I appreciate authors' effort on providing additional experiment results with further explanations to my concerns. I'll remain my score

---

> ### Author Response · Authors · 2024-12-03
> **Response to Reviewer y8Ga**
>
> We thank the reviewer again for all the insightful comments and the time put into helping us to improve our submission.

---

### Official Review · Reviewer_Dfa3 · 2024-11-05

**Soundness:** 3
**Presentation:** 4
**Contribution:** 3
**Rating:** 6
**Confidence:** 3

**Summary:**

The paper proposes QAvatar, a novel CDRL framework that improves data efficiency by combining Q functions from both source and target domains with weight decay functions. Theoretical analysis shows that QAvatar effectively mitigates negative transfer while ensuring reliable transfer. Extensive experiments demonstrate significant improvements in data efficiency and robustness against negative transfer in various RL benchmark tasks.

**Strengths:**

1. QAvatar is a good cross-domain reinforcement learning framework that effectively improves data efficiency , showing great potential for application.

2. The cross-domain Bellman loss function proposed in the paper provides better performance guarantees.

3. The authors conducted extensive experiments, demonstrating that QAvatar achieves excellent results in various environments, indicating high reliability.

**Weaknesses:**

The experimental analysis for the lower quality source section is quite brief. Could you explain why the method performs so well on lower quality sources?

**Questions:**

See questions in Weaknesses.

---

> ### Author Response · Authors · 2024-11-23
> **Rebuttal by Authors**
>
> **Q1: The experimental analysis for the lower quality source section is quite brief. Could you explain why the method performs so well on lower quality sources?**
>
> A1: Thank you for your question. We provide further explanation on the experimental setting and the analysis as follows:
>
> In Figure 4 of the paper, we use a source model that is pre-trained for only 5000 environment steps. In the case of Hopper, the performance of the source model in the source domain is only $220 \pm 32$. Under this condition, the source model could fail to provide positive transfer and could even introduce potential negative effects to the target model's learning (e.g., unseen source $(s,a)$ pairs in the source model, which are randomly initialized, may negatively impact the target model's learning). In such cases, QAvatar effectively mitigates these potential negative effects through the decay function $\alpha(t)$, ultimately achieving a total reward comparable to SAC. More specifically, take the results of Hopper in Figure 4(a) as an example: In the first 200k steps, QAvatar is slightly affected by the low-quality source model. Despite this, QAvatar quickly catches up after 200k steps and even slightly outperforms SAC with the help of the decay function $\alpha(t)$.
> To further strengthen the study on low-quality source models, we further evaluate FT and CAT under the same settings in the Hopper environment.
>
> Results are available at https://imgur.com/a/lompKWY  (and also in Figure F.1 in Appendix F).
>
> From these results, we can observe that QAvatar indeed enjoys favorable transfer capability under both high-quality and low-quality source models compared to FT and CAT.

---

### Official Review · Reviewer_dFmQ · 2024-11-05

**Soundness:** 3
**Presentation:** 2
**Contribution:** 2
**Rating:** 5
**Confidence:** 4

**Summary:**

This paper focuses on cross-domain reinforcement learning and proposes QAvatar to solve transfer across tasks with different state-action spaces. QAvatar includes inter-task mapping learning and combines Q functions from source and target tasks with a proper weight decay function. Experiments are conducted on robotics tasks, compared with several previous transfer methods.

**Strengths:**

1. Learning the inter-task mapping via a Bellman-like loss function looks novel;
2. Both theoretical and empirical analysis are provided.

**Weaknesses:**

1. Although the paper provides theoretical analysis, it heavily depends on previous work.
2. The key implication 2 of Theorem 1 doesn't look original, as most transfer methods include a decay function, that can prevent negative transfer with a large training step t.
3. Some related work should be at least discussed [1-3] in this paper. At least one method with a theoretical guarantee should be compared, for example [3,4] to support the claim in the experiment section.

[1] Measuring the distance between finite markov decision processes. 2016.

[2] A Transfer Approach Using Graph Neural Networks in Deep Reinforcement Learning. 2024.

[3] Cross-Domain Policy Adaptation by Capturing Representation Mismatch. 2024.

[4] Off-Dynamics Reinforcement Learning: Training for Transfer with Domain Classifiers. 2021.

**Questions:**

In Fig 1, it looks like SAC learn from scratch outperforms all other transfer baselines in most of the tasks. The proposed QAvatar is only slightly better than SAC learn from scratch (some of the results are not statistically significant as the errorbars are overlapped). is there any reason for that?

---

> ### Author Response · Authors · 2024-11-23
> **Rebuttal by Authors**
>
> **Q1: Although the paper provides theoretical analysis, it heavily depends on previous work.**
>
> A1: We would like to pinpoint that the theoretical analysis of QAvatar is fundamentally different despite some high-level resemblance with [Zhou et al., 2024]. Below we describe the new techniques of our analysis for CDRL:
> The proof of our Theorem 1 involves two new components that are needed specifically for CDRL:
>
> **(1) Cross-domain Bellman error in Lemma 5:** This is a component specifically needed for CDRL. In Lemma 5, our novelty is two-fold. First, in Lemma 5, we identify the Cross-domain Bellman error as a key component in the bound (cf. Equation (39)). Second, we use a recursive approach to unroll the Cross-domain Bellman error (cf. Equations (40)-(46)) and thereby bound the mean squared error between the Cross-Domain Advantage function $\bar{f}^{(t)}(s,a)$ and the true Target-Domain Advantage function $A^{\pi^{(t)}}$. These are uniquely needed in our CDRL work and do not appear in [Zhou et al., 2024]. Moreover, such **cross-domain Bellman error further motivates the algorithm design of QAvatar** with the cross-domain Bellman loss (cf. Equation (1)).
>
> **(2) Upper bound on Cross-Domain Advantage function (cf. Lemma 2):** While there is seemingly a high-level resemblance between Lemma 2 in our work and Lemma 6 in [Zhou et al., 2024], our Lemma 2 is fundamentally different for addressing cross-domain issues. Specifically, in Lemma 6 of [Zhou et al., 2024], it is shown that they restrict the step size to be small $\eta \leq (1 - \gamma)/2$ and derive a loose constant upper bound for the learning progress of NPG up to \sqrt{t}​. However, in CDRL, this is not desirable since in the scenario of positive transfer, $Q_{src}$​ can well capture the optimal action of $Q_{tar}$​ for most states via inter-domain mappings and hence a larger step size $\eta$ is preferred, allowing the target-domain policy to quickly get close to the optimal one. Based on the above intuition, by contrast, our Lemma 2 avoids the restriction on $\eta \leq (1 - \gamma)/2$ by directly handling the cross-domain advantage function.
>
> [Zhou et al.,2024] Yifei Zhou, Ayush Sekhari, Yuda Song, and Wen Sun. “Offline data enhanced on-policy policy gradient with provable guarantees,” ICLR 2024.
>
> **Q2: The key implication 2 of Theorem 1 doesn't look original, as most transfer methods include a decay function that can prevent negative transfer with a large training step t.**
>
> A2: Thanks for the helpful feedback. While it appears conceptually natural to use a decay function to mitigate negative effects of a source-domain model, **the specific use of a decay function can make a significant difference.** Moreover, we would like to pinpoint that among the baselines considered in our paper, CAT is the only method that also leverages a decay function. Accordingly, we provide a detailed comparison on how CAT and QAvatar utilize the idea of decay function:
>
> **CAT uses a decay function in parameter-based transfer:** CAT is a parameter-based RL transfer method, where the source-domain policy is transferred by taking a weighted linear combination of the hidden layer parameters of the source-domain and target-domain policies. Accordingly, CAT proposed to use a decay term 1-p (where p increases with time) to allow the target model to gradually escape from the negative impact of the source model as the training step t increases. However, parameter-based transfer is known to rely on the assumption that the hidden layers of the source model can capture feature representations that are good to both the source and target domains, and there is no guarantee that this assumption would hold, especially when the domains are fairly dissimilar. This explains why CAT does not achieve effective transfer in most of the tasks in the experiments.
>
> **QAvatar utilizes the decay function in the hybrid Q function:** By contrast, the decay function in QAvatar is to dynamically combine the source-domain and target-domain Q functions. We show that this design can enjoy a nice theoretical upper bound for the sub-optimality gap. Moreover, this design enjoys favorable empirical transfer performance on various tasks, including locomotion, robot arm manipulation, and autonomous driving.
>
> In summary, the use of decay function in QAvatar enables both better empirical cross-domain performance as well as a nice theoretical guarantee.

---

> ### Author Response · Authors · 2024-11-23
> **Rebuttal by Authors**
>
> **Q3: Some related work should be at least discussed [1-3] in this paper. At least one method with a theoretical guarantee should be compared, for example [3,4] to support the claim in the experiment section.**
>
> A3: Thank you for providing the references and the helpful suggestion. We describe the main differences between our work and [1-4] as follows:
>
> **Comparison to [3,4]**: The main difference between [3,4] and our work is the problem formulation. Specifically, [3,4] focus on a cross-domain RL setting which assumes that the source and target domains share the same state space, action space, and even reward function, and the only difference between the two domains is the transition dynamics. By contrast, in our paper, we consider the more general cross-domain RL setting where the state spaces, action spaces, and reward functions can all be different across the two domains. By adopting this setting, QAvatar can enable more flexibility in cross-domain transfer for RL. This distinction in terms of problem setting is also described in the Related Work (Section 2). Moreover, this fact also explains why [3-4] do not need to learn any inter-domain mappings. As all the transfer tasks in our experiments (Section 5) have different state spaces and action spaces across the source and target domains, we did not consider [3,4] as baselines for comparison in the experiments.
>
> **Empirical comparison with PAR [3]:** That being said, if one would like to adapt [3-4] to our general cross-domain RL setting, one direct extension is to employ padding or discarding to address the distinction in the state and action spaces. Below we take the PAR algorithm in [3] as an example as PAR appears to be the SOTA for cross-domain RL problems with identical state space and action space (published in ICML 2024, about two months prior to ICLR submission). In this experiment, we consider two MuJoCo environments (Hopper and Ant) and two Robosuite environments (Door Opening and Table Wiping).
>
> The results are available at https://imgur.com/a/pVGURdR (and also in Figure F.7 in Appendix F).
>
> The results indicate that PAR does not exhibit effective transfer on Ant, Door Opening, and Table Wiping. We conjecture that this is due to the fact that the dimensionalities of the source and target domains are fairly different. This manifests the assumption required by PAR that the state and action spaces need to be identical. The above results demonstrate that [3] is not directly applicable to cross-domain RL under distinct state and action spaces.
>
> **Comparison to [1]:**  The main contribution of [1] is to propose two metrics for measuring the distance between two finite MDPs, under the assumptions that (i) the state representations are equivalent, (ii) the action spaces have a one-to-one correspondence across domains, and (iii) the reward functions are linked in the sense that the rewards are equal when reaching the same subgoals. By contrast, QAvatar is meant to offer a general cross-domain RL algorithm that achieves effective transfer across domains, without measuring the domain distance nor the knowledge about the domain similarity.
>
> **Comparison to [2]:** The main contribution of [2] is to propose TURRET, which focuses on the problem of multi-source transfer by utilizing graph neural networks (GNN) to learn a common state embedding across multiple source-domain tasks. TURRET can be viewed as a variant of CAT with GNN-based state embeddings. By contrast, QAvatar is mainly focused on achieving reliable cross-domain transfer with guarantees without the knowledge about domain similarity, and this problem is already challenging for transfer from one source domain. Despite this, as described in the response to (Q3) for Reviewer y8Ga, we also show that QAvatar can be readily extended to multiple source domains by evaluating QAvatar on a two-source to one-target transfer scenario, where (i) Source domain 1 (referred to as “src1” in the figures below) is Ant-v3 with the front left and back right legs disabled; (ii) Source domain 2 (referred to as ‘src2’ in the figures below) is Ant-v3 with the front right and back left legs disabled; (iii) Target domain (referred to as ‘tar’ in the figures below) is the original Ant-v3 with no modifications. As a preliminary study, we simply assign equal weights to $Q_{src,1}$​ and $Q_{src,2}$ (i.e., $w^{(t)}_1=0.5, w^{(t)}_2=0.5$, for all t), without any hyperparameter tuning.
>
> Results are available at https://imgur.com/a/6X2MVfW (also in Figure F.3 in Appendix F of the updated manuscript).
>
> We also include the above discussion about [1-4] in the related work section (Section 2) of the updated manuscript.

---

> > ### Author Response · Authors · 2024-11-23
> > **Rebuttal by Authors**
> >
> > **Q4: In Fig 1, it looks like SAC learn from scratch outperforms all other transfer baselines in most of the tasks. The proposed QAvatar is only slightly better than SAC learn from scratch (some of the results are not statistically significant as the error bars are overlapped).**
> >
> > A4:
> > - As described in Q1 of Global Response, we provide some reasons why the other transfer baselines (i.e., DCC, CMD, and CAT) perform poorly compared to SAC from scratch. Specifically, DCC and CMD learn in an unsupervised manner and hence do not achieve sufficiently good transfer; on the other hand, the performance of CAT can be attributed to its parameter-based transfer and heuristic nature as well as the slight correlation with its base algorithm.
> > - As described in Q2 of Global Response, through the aggregated Interquartile Mean (IQM), the overall advantage of QAvatar is indeed statistically significant compared to SAC.

---

> > > ### Comment · Reviewer_dFmQ · 2024-11-24
> > > **Thanks for the response**
> > >
> > > Thanks for the clarification.
> > >
> > > One more concern is:
> > >
> > > About the results: I noticed reviewer y8Ga only has this concern and read the authors' response to reviewer y8Ga. The responses don't look convincing. The reviewer checked the CAT paper and found CAT is not only compatible with PPO, but can be combined with any DRL methods. Explaining the benefit of CAT due to PPO is not rigorous. Furthermore, comparing these baselines under different base RL methods looks unfair, I mean, different baselines should use the same base RL algorithm and be fine-tuned suitably.

---

> ### Author Response · Authors · 2024-12-02
> **Response to Reviewer dFmQ**
>
> We sincerely thank the reviewer for the response and for acknowledging our rebuttal addressing the issues. Below we address the follow-up question about the performance and implementation of CAT.
>
> > The reviewer checked the CAT paper and found CAT is not only compatible with PPO, but can be combined with any DRL methods. Explaining the benefit of CAT due to PPO is not rigorous. Furthermore, comparing these baselines under different base RL methods looks unfair, I mean, different baselines should use the same base RL algorithm and be fine-tuned suitably.
>
> Recall from the rebuttal that the major issues with the CAT’s performance are mainly two-fold:
> - **Issue 1:  No guarantees on whether minimizing the Mutual Information loss and the Cycle Consistency loss can always lead to effective transfer.**
> Recall that CAT replies on two major loss terms for cross-domain transfer, namely the Mutual Information loss $L\_{\text{MI}}$ and the Cycle Consistency loss $L\_{\text{cyc}}$. Despite that this is a reasonable design for transfer, there is no theoretical guarantee on whether small $L\_{\text{MI}}$ and $L\_{\text{cyc}}$ is a sufficient condition for effective cross-domain transfer.
>
> - **Issue 2: Parameter-based transfer can be sensitive to domain dissimilarity.**
> CAT is a parameter-based RL transfer method as the source-domain policy is transferred via a weighted linear combination of the hidden layer parameters of the source-domain and target-domain policies. Parameter-based transfer is known to rely on the assumption that the hidden layers of the source model can capture feature representations good for both domains. However, there is no guarantee that this assumption would hold, especially when the domains are fairly dissimilar.
>
> On the other hand, we clearly understand the reviewer’s concern on the potential effect of the base RL algorithm. To address this, we accept the reviewer’s suggestion and try our very best to further implement CAT with SAC as the base RL algorithm (called CAT-SAC). Specifically:
> - Given that there is no open-source implementation of CAT-SAC available, we directly adapt the official implementation of CAT-PPO provided by the original CAT paper [You et al., 2022] by replacing the base algorithm.
> - To ensure the strength of CAT-SAC, we further conduct hyperparameter tuning for the schedule of $p(t)$, which is the weight of the linear combination of hidden layer parameters ($p(t)=0$ means that only the source-domain parameters are used; $p(t)=1$ means that only the target-domain parameters are used). Specifically, [You et al., 2022] sets $p(t)$ to be piecewise linear as follows: Let $T$ be the total training steps. (i) $p(t)=0$, for $t\in [0,  c\_1 T]$. (ii) $p(t)$ grows linearly from 0 to 1 during $t\in [c\_1  T, c\_2 T]$, (iii) $p(t)=1$, for $t\in [c\_2 T, T]$. In the official CAT-PPO, [You et al., 2022] chooses $c\_1=0.45, c\_2=0.9$. Here, for each task, we choose the best among the following candidate choices: $(c\_1,c\_2) \in \\{(0.25, 0.6), (0.35,0.7), (0.45, 0.9)\\}$.
>
>
> The experimental results are available as follows:
> - Evaluation rewards: https://imgur.com/a/TExHOZo
> - $L\_{\text{MI}}$ and $L\_{\text{cyc}}$: https://imgur.com/a/cEoH77m
>
> **Remark**: During training, $L\_{\text{cyc}}$ shall be made as close to 0 as possible by definition. Regarding $L\_{\text{MI}}$, as there is no clear lower bound of $L\_{\text{MI}}$, we also report the minimum $L\_{\text{MI}}$ obtained under CAT-PPO during transfer for each task in the figures above as a reference for comparison.
>
>  Based on the above results, we highlight several key observations:
> - **We observe that CAT-SAC can suffer from Issue 1**. Notably, CAT-SAC can have small $L\_{\text{MI}}$ and $L_{cyc}\approx 0$ in all the tasks. However, small $L\_{\text{MI}}$ and $L\_{\text{cyc}}\approx 0$ do not necessarily lead to effective transfer.
>     - For example, in all the three Robosuite tasks (i.e., Block Lifting, Door Opening, and Table Wiping), while $L\_{\text{MI}}$ and $L\_{\text{cyc}}$ of CAT-SAC are all sufficiently low, the transfer is not effective.
>     - On the other hand,  the initial transfer for Ant is quite effective and both $L_{MI}$ and $L\_{\text{cyc}}$ are favorably low.
> - **Issue 2 can also occur under CAT-SAC**. Specifically, based on the results on Ant, we can see that the performance is fairly unstable during the time when $p(t)$ grows from 0 to 1 (with $c_1=0.35$, $c_2=0.7$). This is exactly the phase when the hidden layer parameters from both domains are linearly added. This demonstrates the fundamental issue with parameter-based transfer.
>
> We will add the above experimental results to the final version of the paper to make the comparison more comprehensive. Again, we thank the reviewer for all the detailed review and the time the reviewer put into helping us improve our submission.

---

### Author Response · Authors · 2024-11-23
**Global Response (Part 1)**

We greatly thank all the reviewers for the valuable feedback and insightful suggestions. Here we first provide our response to the questions asked by multiple reviewers, and the responses to the rest of the questions are posted below each individual review.

**Q1: Explain why most of the baselines are poor compared to SAC.**

A1: **Regarding DCC and CMD**: The scores reported in the original papers are indeed much lower than those of SAC, as summarized in the sanity check for reproduction in Section C.2 of Appendix C of the original manuscript (also see the table below).
The main reason for the poor performance of DCC and CMD is that they learn in an unsupervised manner and hence do not utilize any information about the target rewards during training.
|             | DCC (original) | DCC (reproduced) | CMD (original) | CMD (reproduced) | SAC             | QAvatar (Ours)  |
| ----------- | -------------- | ---------------- | -------------- | ---------------- | --------------- | --------------- |
| Swimmer     | 204 $\pm$ 56   | 132 $\pm$ 47     | 44 $\pm$ 38    | 41 $\pm$ 14      | 235 $\pm$ 141   | 316 $\pm$ 139   |
| Halfcheetah | 2471 $\pm$ 382 | 1360 $\pm$ 729   | 2114 $\pm$ 332 | 303 $\pm$ 75     | 11445 $\pm$ 1897 | 12819 $\pm$ 679 |
| Ant         | N/A             | 973 $\pm$ 501    | 649 $\pm$ 347  | 882 $\pm$ 52     | 2290 $\pm$ 785  | 2840 $\pm$ 1532 |

Remark: Regarding DCC, we directly use the official implementation provided by the original DCC paper. Moreover, as there is no CMD implementation available, we reproduced CMD by referring to the source code of DCC as these two algorithms are similar. We also follow the same hyperparameter configurations as listed in the CMD paper. With that said, we have tried our best in reproducing the baseline results.

**Regarding CAT**: The issues with its performance are mainly three-fold:

(1) **Parameter-based transfer**: CAT can be viewed as a parameter-based RL transfer method as the source-domain policy is transferred by taking a weighted linear combination of the hidden layer parameters of the source-domain and target-domain policies. In the general context of transfer learning, parameter-based transfer is known to rely on the assumption that the hidden layers of the source model can capture feature representations that are good to both the source and target domains (e.g., see [Zhuang et al., 2021]). However, there is no guarantee that this assumption would hold, especially when the domains are fairly dissimilar. Moreover, such direct neural network parameter sharing can be rather sensitive to the magnitude of the hidden layer parameters across the source and target models.

(2) **No theoretical guarantees**:  Recall that CAT replies on two major loss terms for cross-domain transfer, namely the Mutual Information loss $L\_{\text{MI}}$ and the Cycle Consistency loss $L\_{\text{cyc}}$. However, there is no theoretical guarantee on whether small $L\_{\text{MI}}$ and $L_{\text{cyc}}$ is a sufficient condition for effective cross-domain transfer. Moreover, based on (1), without additional assumptions, there is no clear theoretical connection between the performance of the source-domain policy and that of the weighted linear combination in the parameter space. Therefore, this also motivates the need for an algorithm with theoretical grounding like QAvatar.

(3) **Potential effect of the base algorithm**: Given that CAT is fairly sophisticated and involves many components (cf. Figure 1 of the CAT paper), we use the official implementation provided by the CAT paper to ensure the correctness and reproducibility. CAT is by default a PPO-based algorithm, and we conjecture that its performance can be slightly correlated with the performance of vanilla PPO.  We provide the learning curves for four environments (i.e., HalfCheetah, Hopper, Door Opening, and Table Wiping) at https://imgur.com/a/347qFho and also in Figure F.6 in Appendix F. Specifically:
* Door Opening: Both CAT and PPO perform poorly.
* Hopper and Halfcheetah: While PPO performs reasonably well, CAT still suffers.
* Table Wiping: Both CAT and PPO perform quite well.
This corroborates the slight correlation between CAT and its base algorithm PPO.

[Zhuang et al., 2021] Fuzhen Zhuang, Zhiyuan Qi, Keyu Duan, Dongbo Xi, Yongchun Zhu, abd Hengshu Zhu, “A Comprehensive Survey on Transfer Learning,” Proceedings of the IEEE, 2021.

---

### Author Response · Authors · 2024-11-23
**Global Response (Part 2)**

**Q2: Discuss the statistical significance of QAvatar’s performance.**

A2:  The overall advantage of QAvatar is indeed statistically significant compared to SAC: To provide a more reliable statistical analysis of QAvatar’s performance, we follow the general methodology proposed by [Agarwal et al., 2021] and calculate the Interquartile Mean (IQM) using rliable (https://github.com/google-research/rliable). This approach enables better evaluation of results at an aggregated level. Due to the varying reward ranges in different environments, as suggested by [Agarwal et al., 2021], we normalized the total reward to the range [0,1] in each environment to allow comparisons across environments.

IQM: https://imgur.com/a/65TrGGM (also in Figure F.2 in Appendix F)

The above results show the aggregated IQMs for all environments (including Locomotion, Robot Arm Manipulation, and Autonomous Driving), along with the 95% stratified bootstrap confidence intervals. We observe that QAvatar indeed achieves significantly better performance than all the baselines.

[Agarwal et al., 2021] R. Agarwal, R., M. Schwarzer, P. S. Castro, A. C. Courville, and M. Bellemare, “Deep reinforcement learning at the edge of the statistical precipice,” NeurIPS 2021.

---

### Meta-Review · Area_Chair_mfZN · 2024-12-21

**Metareview:**

(a) This paper proposes QAvatar, a cross-domain reinforcement learning framework that addresses the challenge of knowledge transfer between domains with distinct state and action spaces. QAvatar leverages a weighted combination of a learned target-domain Q function and a pre-trained source-domain Q function to improve sample efficiency while mitigating negative transfer. The proposed hybrid Q function approach combines information from both domains and uses a decay function to adjust the reliance on the source domain over time. The paper provides theoretical analysis establishing the convergence properties of QAvatar. The experimental results showcase QAvatar’s ability to accelerate learning in the target domain, outperform baseline methods (including those learning from scratch), and maintain robustness even when the source domain model is of low quality or introduces negative transfer.

(b) Strengths:
- The paper provides both theoretical guarantees for the convergence of QAvatar and extensive empirical validation across diverse environments, demonstrating the practical benefits of the proposed approach.
- Robustness and mitigation of negative transfer: QAvatar incorporates a decay function that allows it to adapt to scenarios where the source domain model is of low quality or might lead to negative transfer, enhancing its reliability and practical applicability.
- The paper provides clear explanations of the methodology, theoretical analysis, and experimental setup.

(c) Weaknesses:
- The current study focuses on transfer from a single source domain, instead of a generalist generalization setting.

(d) The decision is to reject. There is no obvious strengths of the paper. All the reviewers are on the borderline (5, 6, 6, 6). I think the impact of transferring from single source domain to single target domain with limited evaluation in sim is perhaps too little impact to the current community -- which seeks for impacts that are realistic, scalable, and generalizable.

**Additional Comments On Reviewer Discussion:**

During the rebuttal period, several issues were raised by reviewers. In particular:
- Reviewers dFmQ, qMj1, y8Ga questioned the seemingly insignificant performance margin between QAvatar and SAC, a baseline that learns from scratch. They noted that QAvatar, while showing slight improvement in data efficiency, often requires a similar training time to SAC, raising concerns about the claimed improvement. The authors addressed this by clarifying the metrics for evaluating data efficiency.
- Reviewer qMj1 questioned the benefit of cross-domain transfer when the source-domain model is of low quality. They argued that, in such cases, training directly in the target domain from scratch might be more practical. The authors acknowledged this point and emphasized that QAvatar remains reliable even under negative transfer, a crucial property in practical scenarios where pre-determining the transfer type (positive or negative) can be challenging.
- Reviewers dFmQ, qMj1, y8Ga sought clarification on the performance of QAvatar in negative transfer scenarios and specific situations where transfer might fail. The authors presented additional experimental results involving negative transfer scenarios in various environments to address.

Although the discussion improves the paper, I think the impact of this paper is still too limited. All reviewers don't feel very strong about the paper.

---

### Decision · Program_Chairs · 2025-01-22

Reject